# SortBench: Benchmarking LLMs based on their ability to sort lists

## Abstract

Sorting is a tedious but simple task for human intelligence and can be solved fairly easily algorithmically. However, for Large Language Models (LLMs) this task is surprisingly hard, as some properties of sorting are among known weaknesses of LLMs: being faithful to the input data, logical comparisons between values, and strictly differentiating between syntax (used for sorting) and semantics (typically learned by embeddings). Within this paper, we describe the new SortBench benchmark for LLMs that comes with different difficulties and that can be easily scaled in terms of difficulty. We apply this benchmark to seven state-of-the-art LLMs, including current test-time reasoning models. Our results show that while the o3-mini model is very capable at sorting in general, even this can be fooled if strings are defined to mix syntactical and semantical aspects, e.g., by asking to sort numbers written-out as word. Furthermore, all models have problems with the faithfulness to the input of long lists, i.e., they drop items and add new ones. Our results also show that test-time reasoning has a tendency to overthink problems which leads to performance degradation. Finally, models without test-time reasoning like GPT-4o are not much worse than reasoning models.

## 1 Introduction

Sorting is a basic skill that humans acquire at an early age already: we learn the order of numbers, the alphabet, and we also learn to sort by shapes or similar aspects. While sorting with very large numbers of items is a tedious activity for humans, we can solve it reliably nonetheless. Moreover, algorithmic sorting strategies are well-known and range from intuitive (bubble sort, insertion sort) to more optimized variants (quicksort, mergesort). Thus, from both a human intelligence perspective as well as from the algorithmic complexity required to solve the problem, sorting is a simple problem.

Modern Large Language Models (LLMs) based on decoder-only transformers (Radford et al., 2018), instruction fine-tuning (Zhang et al., 2023), and even advanced chain-of-thought reasoning (Jaech et al., 2024) are showing increasingly strong capabilities at even solving hard problems like advanced math (AoPS Online, 2024) or the ARC challenge (Price, 2024). Still, we believe that the simple task of sorting a list is still hard and yet unsolved for such models. Concretely, we believe that sorting has several properties that make this both a hard challenge for LLMs and a good benchmark to understand important LLM qualities like faithfulness to the input, logical comparisons, and the capability to strictly differentiate between the syntax of the input (relevant for sorting) and its semantics (typically learned by embeddings).

The idea of SortBench is simple: use a randomized generator to create lists, give these lists to an LLM with a prompt that instructs it to sort them, and then evaluate if the lists were sorted correctly and returned in the expected output format. The evaluation considers whether the order of list items is correct and if the items in the returned list are the same as in the unsorted list. Using different generators for the lists allows us to evaluate different properties, e.g., sorting of integers, floats, random strings, and words for lists of different lengths. The difficulty can be further increased by creating variants of the above, e.g., requiring a lexicographic sorting of words that represent numbers (e.g., one, two, three). The benchmark is designed to assess both the general sorting capabilities of LLMs and also to understand which properties of data

make sorting more difficult for them, whether they follow the instructions of the prompt, and whether they manage to leave the content of the list unchanged.

The relatively simple design that covers the analysis of a simple task in depth is different from other benchmarks focused on algorithmic reasoning like the CLRS-Text (Markeeva et al., 2024): instead of considering if LLMs can conduct reasoning in a sequence of steps required by certain algorithms (e.g., quick sort, bubble sort), we only assess if the LLMs can sort at all. This is better comparable with the arithmetic tasks of MathBench (Liu et al., 2024), which also do not consider specific algorithms for arithmetic, but rather only assess if LLMs can perform arithmetic tasks. However, in contrast to both benchmarks, we also consider the difficulty of the tasks for different data as well as a gradual, non-binary assessment of the ability to solve tasks. While this increases the complexity of the benchmark for this task, this allows us to understand properties beyond sorting, i.e., how different data types and input lengths affect the results.

## 2    Method

Below, we describe the generation of the SortBench v1.0 benchmark for the basic and advanced difficulty, as well as debug tasks that are useful to better understand the LLM's capabilities. SortBench focuses on lists over two input domains, i.e., sorting numbers written in decimal notation and lexicographic sorting of strings. For the strings, we do not consider special characters, but rather restrict the benchmark to ASCII letters and non-English words. Further, we write all lists in the format as Python lists, i.e., with opening and closing brackets `[]`, numbers without quotation marks and strings within quotation marks.

### 2.1    Prompt

Our benchmark is designed to evaluate the zero-shot capabilities of LLMs without any fine-tuning neither through modifying weights nor through prompt engineering. While specifically crafted and engineered prompts, including few-shot prompts, may lead to better results, a basic capability like sorting should not require such considerations. Hence, we only give the models a simple, straightforward prompt telling them what to do, without any details on the exact, lengthy specifications or data formats, or similar. We use the following system and user prompts:

- System prompt: "Your task is to sort a list according to the common sorting of the used data type in Python. The output must only contain the sorted list and nothing else. The format of the list must stay the same."

- User prompt: "Sort the following list: <list>" where <list> is replaced with the list that needs to be sorted in Python, e.g., `[16, 5, 10]` for an integer list or `['foo', 'bar', 'baz']` for a list of strings.

### 2.2    Benchmark tasks

The tasks within our benchmark are grouped by their difficulty. We have basic tasks, that should align well with how LLMs are trained, advanced tasks that rather push the LLMs to cases that might be problematic, and debug tasks that can be used for additional diagnostics with respect to how well LLMs follow the general instruction of sorting a list without changes to the list's items.

#### 2.2.1    Basic difficulty

For the basic difficulty, we generate lists that are simple in the sense that their contents should align fairly well with how LLMs are trained, specifically, list items they likely see often during the training. This is based on the assumption that logical comparisons are easier for LLMs between tokens that were seen during training. Further, we avoid aspects that might confuse the sorting, e.g., having the same list item multiple times or using negative numbers. Thus, the basic difficulty evaluates the general capability to follow the instruction to sort a list, since there are no possible confusing constructs in the tasks and all list items are from the same input domain (i.e., numbers or words). Based on these goals, we use the following three types of lists for the basic difficulty:

- **Int-0:1000**: Integers in the range 0 to 1,000 without duplicates.

- **Float-0:1000**: 32-bit floating point numbers printed a full precision decimal notation in the range 0 to 1,000 without duplicates.

- **English**: Words from the English language sampled from the WordNet corpus (Fellbaum, 2010) without duplicates. Further, we disallow words with an apostrophe to avoid potential issues with with Python's string parsing.

### 2.2.2 Advanced difficulty

With the advanced difficulty we want to understand if orders are not only memorized for token combinations that may have been known from training, but also to other, more complex, scenarios. Note that we do not consider duplication of list items as part of this difficulty, as we rather consider this as a debug task (see Section 2.2.3). For numbers, we make the task more difficult by looking at uncommonly large values and values that are very close to each other, i.e., compressed within a small interval. Further, we add lists with negative values to measure whether signs are interpreted correctly by the LLMs. This results in the following advanced tasks for numbers:

- **Int-10000000:10001000**: Integers in the range 10,000,000 to 10,001,000 without duplicates.

- **Float-10000000:10001000**: 32-bit floating point numbers printed at full precision in decimal notation in the range 10,000,000 to 10,001,000 without duplicates.

- **Float-0:0.0001**: 32-bit floating point numbers printed at full precision in decimal notation in the range 0 to 0.0001 without duplicates.

- **Int-n1000:1000**: Integers in the range -1,000 to 1,000 without duplicates.

- **Float-n1000:1000**: 32-bit floating point numbers in the range -1,000 to 1,000 without duplicates.

For strings, the easiest way to increase the difficulty is to not use actual words, but rather random strings. However, we can also try to actively confuse the model. For this, we consider two scenarios. First, we add a prefix of equal characters (e.g., "rrrHello"'). This requires the sorting to basically ignore the beginning of the tokens associated with a list item and sort based on later characters in the string. Second, we try to confuse the LLM by using number words (e.g., "one", "three-thousand"). Through this, we try to trick the LLM as it may rather consider the semantic meaning of the number words and sort by the numbers instead of the lexicographic order. This results in the following advanced tasks for strings:

- **ascii**: Strings of five random, lowercase ASCII letters without duplicates

- **AsCiI**: Strings of five random ASCII letters with both lower and uppercase letters without duplicates.

- **PrfxEnglish**: Words from the English language without duplicates that have a constant letter repeated three times as prefix. The prefix is the same for all words in a list.

- **NumberWords**: Words that represent integers between one and 1,000 without duplicates.

### 2.2.3 Debug tasks

In addition to the tasks above used for scoring LLMs, we also include two types of debug tasks that support the diagnostics of LLMs with respect to their faithfulness to the inputs. Through these, we want to enable further insights into the benchmark scores by providing comparisons to scenarios which require the LLMs to respect properties of the input, independent of sorting. For this, we provide two additional variants for all three tasks from the basic difficulty, i.e., six tasks. In the first variant, the lists are already sorted, which reduces the task of the LLM to just repeat a list. This helps us to see the if LLMs are able to not change the

lists in isolation, i.e., regardless of mistakes that may be induced here due to sorting. In the second variant, each list item is duplicated, i.e., appears twice in the list. This requires the LLM to not just recognize the word order, but also repeat words the required number of times. This helps us to understand if the LLMs treat list items as separate entities. Bad performance here indicates that syntactic (and possibly also semantic) similarity between list items may affect the task.

## 2.3 List lengths

We create for each of the above tasks of eight lists different lengths using an exponential growth with a base of two, such that we have lists with $2^1 = 2, ..., 2^8 = 256$ items. For each length, we create 10 lists to account for possible random effects. Since we have three basic tasks, nine advanced tasks, and six debug tasks, this means we have to sort $8 \cdot 10 \cdot (3 + 9 + 6) = 1440$ lists. The maximum list length was selected such that typical tokenizers based on Byte Pair Encoding (BPE, Sennrich et al., 2015) or similar have less than or equal to 4096 tokens, incl. the system and user prompt. This ensures that the sorting could happen in a context window of 8192 tokens.

## 2.4 Scores for LLM-Sorting

When we consider the quality of sorting with LLMs, there are three general aspects to consider, i.e., whether the output is a valid Python list, whether the list items match the original list, and whether the result is correctly sorted. For all aspects, we design metrics that enable us not only to judge if the property is in a binary manner fulfilled (i.e., valid or invalid, list items unchanged or changed, and sorted or not fully sorted), but rather in a gradual manner that enables us to judge how close models are at fulfilling a certain property. This allows us to gain deeper insights into the differences between models and their ability to produce outputs that fulfill the desired properties.

### 2.4.1 Output validity

The first challenge when working with LLMs is to ensure that their output follows an expected specification. In our case, this means that the output list is still a valid Python list. To check this, we use Python's `eval` function that takes as input a string and interprets this as a Python command. With a valid list, this would be directly converted. If that fails, we check if the output generation was possibly prematurely aborted, i.e., the LLM yielded an end-of-sequence token when the list was not yet finished. For that, we first check if the output finishes with a closing bracket. If that is not the case, we manually check the list and look for alternative ways to parse the list. Section 3.7 presents a detailed list of error cases we found, including their frequency. After parsing, we additionally check if the list contains an ellipsis (...), because this is a special data type in Python, cast all list items to the expected type (e.g., '100' to int), and check if the output was parsable, though not a list, but rather a tuple. Based on these outcomes, we define the $ValidityScore$ as follows:

- $ValidityScore = 1$, if the output is a valid Python list, without ellipsis and of the correct type.

- $ValidityScore = 0.75$ if the output is valid Python, but contains an ellipsis; the type of the list items needed to be cast; the output was a tuple instead of a list; or if the output was only missing the closing bracket.

- $ValidityScore = 0.5$, if the output contained a list, but this required special parsing (see Section 3.7).

- $ValidityScore = 0$ if the output did not contain a parsable list.

### 2.4.2 Sorting correctness

We use two metrics for the correctness of the sorting that would be zero for correctly sorted lists:

- $UP$ is the percentage of unordered pairs. We conduct pair-wise comparisons between all list items and compute the ratio of such pairs that are not in order.

- $UN$ is the percentage of unordered neighbors. For this, we compute the ratio of directly adjacent list items that are not in order.

With $UP$, we get a global perspective that penalizes the sorting if elements are far off from their current position. Such items will be part of many unordered pairs, increasing the count. With $UN$, we get a local perspective and rather consider if most items have a suitable neighbor, such that a single item in a wrong position has a low impact. We use the mean of these two metrics to define

$$SortingScore = 1 - \frac{UP + UN}{2} \tag{1}$$

for a list.

### 2.4.3 Faithfulness

All metrics considered so far are solely computed based on the output of the LLM, i.e., the result of the sorting, without taking the list that was supposed to be sorted into consideration. However, there are no guarantees that the returned lists actually have the same items as the input: LLMs are known to hallucinate, which in this case means altering the list by either dropping or adding items. To account for this, we define two additional metrics:

- $I^+$ is the ratio of added list items, i.e., the number of items that were added in relation to the length of the original list.

- $I^-$ is the ratio of the missing items, i.e., the number of items that were present in input but not in the original list in relation to the length of the original lists.

Please note that both metrics are computed accounting for duplicates, i.e., if an item appears twice in the input, it also needs to appear twice in the output. Further, we clip $I^+$ at one to avoid possible cases in which a LLM might add more new items, than were originally in the list. Same as above, we use the mean of both metrics to define

$$FaithfulnessScore = 1 - \frac{I^+ + I^-}{2}. \tag{2}$$

### 2.4.4 Total score

The total score for a list is defined using the three criteria validity, sorting correctness, and faithfulness as

$$SortBenchScore = ValidityScore \cdot \frac{SortingScore + FaithfulnessScore}{2}. \tag{3}$$

Hence, we compute the mean of the sorting and faithfulness scores and penalize this, if the list cannot be parsed.

All scores so far were computed per list, i.e., for each list in every task. While these scores allow good insights into different strengths and weaknesses of the LLMs, benchmarks should ideally have a single value per model per task as final result for ranking. We compute this using a length-weighted mean over all scores for a task:

$$ModelScore = \frac{\sum Length \cdot mean(SortBenchScore_{Length})}{\sum Length}. \tag{4}$$

The weighting by length ensures that all sequence lengths in the benchmark are considered, while preventing very short, easier-to-sort sequences from dominating the total score. This is important because of the exponential growth of lengths, which means we have more lists with short length than with longer lengths. An alternative interpretation for this definition is that it is the area under the scoring curve with the sequence length on the x-axis and the score on the y-axis. These scores can then be averaged to get the performance per difficulty, as well as overall.

## 2.5 Model selection

We used SortBench to evaluate a broad range of LLMs. We include current state-of-the art models without test-time reasoning,[1] i.e., GPT-4o and Claude Sonnet 3.5, as well as their smaller counterparts GPT-4o-mini and Claude 3.5 Haiku (Hurst et al., 2024; Anthropic, 2024). We use LLAMA-3.1-70b (Grattafiori et al., 2024) to represent open-weights models without test-time reasoning. We use o3-mini (OpenAI, 2025) as proprietary and DeepSeek-r1-70b (DeepSeek-AI, 2025) as open-weights models that support test-time reasoning. From a scientific point of view, DeepSeek-r1-70b is preferable, because we have access to the reasoning tokens. However, the costs of hosting DeepSeek-r1-70b locally are very high, because the reasoning processes can become very long, meaning that the outputs were several times larger than expected for only sorting. Though we cannot access the reasoning tokens, we also include the proprietary o3-mini model from OpenAI in our benchmark. The pricing for this model is reasonable and comparable to GPT-4o. Moreover, we can at least get access to the summary of the reasoning by using the ChatGPT web interface (see Section 3.1). We did not include more reasoning models due to the prohibitive costs involved, e.g., OpenAIs o1, the more expensive, larger cousin o1 of the o3-mini model.

An important requirement for this model selection was that the whole sorting can happen within the context window of the model, which is the case for all of the above models. This avoids aspects like how longer contexts are supported to influence the results. Unfortunately, this is not always the case, as, e.g., the Gemma (Gemma Team et al., 2024), Gemini (Google, 2024), and Qwen 2.5 (Yang et al., 2024) model families tokenize numbers through their digits. Due to this, our advanced tasks with longer numbers were too long to fit within the context window.

# 3 Results

Table 1 shows the scores for the benchmark. Visualizations of the results for different list lengths for all tasks can be found in Appendix A.1. We report the results of statistical tests conducted following the guidelines from Benavoli et al. (2017) for all list lengths in Appendix A.1. All implementations we created for this work are publicly available online: BLINDED

## 3.1 o3-mini performance dominates

The o3-mini test-time reasoning model from OpenAI outperforms the other models in the benchmark: it almost always follows the instruction to output only a Python list, is most faithful to the input, and yields the best sorted list. For the *SortingScore* the difference to Claude-3.5-Sonnet is extremely small, with a very small advantage for Claude-3.5-Sonnet for the basic and debug tasks and a small advantage for o3-mini on the advanced tasks. For all scores, GPT-4o is typically a bit worse than the o3-mini model, but the difference is not very large. In the debug tasks, GPT-4o even slightly outperforms o3-mini due to a better faithfulness. As we discuss in more detail in Section 3.6, the difference to the best other model (GPT-4o) is only significant for the longest lists with 256 items.

Looking at the results in greater detail using the data reported in the appendix, o3-mini *almost* solves the task at hand, i.e., is almost able to sort reliably. There are two open problems: First, o3-mini gets fooled by the NumberWords, where we observe problems with the *FaithfullnessScore*, while the *SortingScore* remains very high. A look at the data shows the reason for this: the model often converts the strings into the corresponding integers and then sorts these. The output is, therefore, a sorted list but not with the expected, original list items. Another perspective is that the model got distracted from the task that only required looking at the syntax by instead using the semantics of the list items. We note that models without test-time reasoning do not have this problem. The second problem is also with the *FaithfullnessScore*, which drops for the longer lists of the Float-0:0001, ascii, AsCiI, and PrfxEnglish advanced tasks. The same happens for the English-Duplicate debug task. All these tasks share the common trait that they require a comparably larger number of tokens for the list items. Thus, maintaining faithfulness to the input seems to become problematic as the context length increases.

---

[1]i.e., a variant of test-time compute efforts used for reasoning and planing Ji et al. (2025)

| | Model | $ModelScore$ | $SortingScore$ | $FaithfulnessScore$ | $ValidityScore$ |
|---|---|---|---|---|---|
| **Basic** | o3-mini | **0.984** | 0.997 | **0.970** | **1.000** |
| | GPT-4o | 0.931 | 0.986 | 0.955 | 0.954 |
| | GPT-4o-mini | 0.901 | 0.978 | 0.888 | 0.964 |
| | Claude-3.5-Sonnet | 0.862 | **0.998** | 0.912 | 0.895 |
| | Claude-3.5-Haiku | 0.830 | 0.992 | 0.904 | 0.869 |
| | Llama-3.1 | 0.727 | 0.989 | 0.709 | 0.832 |
| | DeepSeek-r1 | 0.562 | 0.823 | 0.694 | 0.679 |
| **Advanced** | o3-mini | **0.963** | **0.982** | **0.946** | **0.999** |
| | GPT-4o | 0.852 | 0.885 | 0.889 | 0.953 |
| | GPT-4o-mini | 0.758 | 0.851 | 0.744 | 0.945 |
| | Claude-3.5-Sonnet | 0.775 | 0.925 | 0.870 | 0.854 |
| | Claude-3.5-Haiku | 0.750 | 0.892 | 0.850 | 0.852 |
| | Llama-3.1 | 0.645 | 0.843 | 0.658 | 0.836 |
| | DeepSeek-r1 | 0.450 | 0.836 | 0.614 | 0.593 |
| **Debug** | o3-mini | 0.995 | 0.999 | 0.991 | **1.000** |
| | GPT-4o | **0.998** | 0.998 | **0.998** | **1.000** |
| | GPT-4o-mini | 0.938 | 0.990 | 0.905 | 0.990 |
| | Claude-3.5-Sonnet | 0.863 | **1.000** | 0.913 | 0.895 |
| | Claude-3.5-Haiku | 0.844 | 0.998 | 0.898 | 0.886 |
| | Llama-3.1 | 0.780 | 0.980 | 0.803 | 0.858 |
| | DeepSeek-r1 | 0.615 | 0.991 | 0.756 | 0.684 |
| **All tasks** | o3-mini | **0.977** | **0.990** | **0.965** | **0.999** |
| | GPT-4o | 0.914 | 0.940 | 0.937 | 0.969 |
| | GPT-4o-mini | 0.842 | 0.919 | 0.823 | 0.963 |
| | Claude-3.5-Sonnet | 0.819 | 0.962 | 0.891 | 0.875 |
| | Claude-3.5-Haiku | 0.795 | 0.944 | 0.875 | 0.866 |
| | Llama-3.1 | 0.704 | 0.915 | 0.716 | 0.843 |
| | DeepSeek-r1 | 0.524 | 0.887 | 0.677 | 0.638 |

Table 1: Results sorted by the $ModelScore$ in all tasks, reported averaged over the different sets of of tasks (basic, advanced, debug) and overall.

To get insights into the reasoning of o3-mini, we used the ChatGPT Web frontend to sort a single list with 256 items for each task. While this does not give us access to the reasoning tokens, we can at least access a generated summary. The first reasoning step was always a repetition of the task at hand, that extended our prompt with a correct analysis of the data type and expected sorting order of the list. For about half of the tasks, it stopped there. In all these cases, the result was a correctly sorted list. In other cases, the model complained that using Python's `sorted` would be more convenient and that it needs to be "extra careful". We ran the NumberWords two times: in one case, the model stuck to lexicographic sorting and ignored that the strings represented numbers, in the second case the model did was we described above, i.e., it identified that the strings represent numbers, converted them, and sorted them as integers. For one of the examples we tried, i.e., the basic English task (see Appendix A.7), the reasoning process was very long, considered many individual words and characters, and (the summary provided on the Web page) even switched languages twice: the reasoning started in English, then switched to German, and then back to English. Whether these language switches are an artifact of the summarization or actually happened during the reasoning is unclear. Perhaps not surprisingly, this led to the worst performance among all lists we sorted using the Web-frontend. Overall, this qualitative analysis of the reasoning process is in line with the quantitative results from the benchmarks and shows that reasoning, in general, helps the model to elaborate on the prompt to make the result better, but also has an inherent risk that the model starts to *overthink* causing hallucinations.

That such overthinking occurs was also already studied by others. Notably, Chen et al. (2024a) found that test-time reasoning models required almost 20 times the number of tokens for simple mathematical tasks than models without test-time reasoning. Our analysis adds to these findings that it is not only often expensive, it may also actually degrade performance sometimes. While this did not happen very often with o3-mini, overthinking was a very large problem for DeepSeek-r1 (see Section 3.5).

## 3.2 Large, proprietary models are very good, even without test-time reasoning

The other large, proprietary models in our benchmark, GPT-4o and Claude-3.5-Sonnet, also performs well and our statistical analysis shows that the difference to o3-mini only becomes relevant at list length of 128 for Claude-3-5-Sonnet and 256 for GPT-4o.

GPT-4o is a close second to o3-mini and generally outperforms the other models, based on the *ModelScore*. It consistently outperforms all models except o3-mini in terms of faithfulness, i.e., not changing the list items and also performs best in the advanced and debug tasks when it comes to the validity. Only for the basic task, the validity of GPT-4o-mini is slightly higher. This indicates that GPT-4o is very good at following the instruction that the output should only be a Python list and also that there is a relatively low rate of hallucinations in this setting, same as the reasoning counterpart o3-mini from OpenAI. The sorting capabilities of GPT-4o are also strong, though there are already a couple of mistakes for the longer lists in the basic task. The advanced tasks reveal a clear weakness for the sorting of strings in more complex setting (ascii, AsCiI, PrfxEnglish, and NumberWords), none of which works reliably for longer sequences. Moreover, the sorting of lists that include negative numbers (Int-n1000:1000) is also a problem for GPT-4o. The other advanced tasks, including the sorting of small ranges of floats, work well. The debug tasks show that the model also does not get easily tripped up by already sorted lists or duplicates within the lists.

Claude-3.5-Sonnet is a bit better at sorting than GPT-4o, as shown by the better *SortingScore* for all three groups of tasks. The sorting is perfect for the debug tasks and almost perfect for the basic tasks, i.e., the sorting capabilities are similar to those of o3-mini. For the advanced tasks, there are smaller problems for all tasks with longer list, but overall fewer bad sortings than with GPT-4o. Notably, the model handles both adversarial string tasks (PrfxEnglish, NumberWords) as well as lists with negative values (Int-n1000:1000) very well. However, the validity and faithfulness are both worse than for GPT-4o. The validity mostly suffers from not cleanly closing lists (see Section 3.7). However, the lower faithfulness shows that the model has problems with keeping the list items as is. Claude-3.5-Sonnet has problems with the faithfulness for all string-based and float-based tasks, regardless of whether they are basic, advanced or for debugging. Only integers are reliably unchanged during the sorting.

### 3.3 Smaller proprietary models have the same properties as their large brothers

The smaller variants of the proprietary models, GPT-4o-mini and Claude-3.5-Haiku have similar strengths and weaknesses than their larger counterparts, i.e., GPT-4o-mini is better when it comes to faithfulness and validity than Claude-3.5-Haiku, while Claude-3.5-Haiku has better *SortingScore*s. The gap between GPT-4o-mini and GPT-4o is fairly large for the advanced tasks, as both the sorting and faithfulness strongly suffer across all tasks. However, for the simpler basic and debug tasks, GPT-4o mini is performing very strongly, actually outperforming even the larger Claude-3.5-Sonnet model due to the higher validity. The gap between Claude-3.5-Sonnet and Claude-3.5-Haiku is more consistent, i.e., there is a relatively small gap across all metrics and task.

### 3.4 LLAMA has problems with long lists

Overall, LLAMA-3.1 performs quite well in the benchmark: the performance is overall comparable to that of GPT-4o-mini and Claude-3.5-Haiku, which is expected given that we use the LLAMA variant with 70B parameters. When we analyze the performance of LLAMA in depth, we observe that long lists with 256 items are the reason, why the model does not score better. For all tasks, we observe a sharp drop in performance for these longer lists. This indicates that while the lists (incl. the sorted list) fit into the context window, the model obviously still has problems capturing relationships that span almost the whole context window. Further supporting this is that this drop in faithfulness and validity already happens earlier for the lists with floating point numbers and random strings as items: these lists require the most tokens among all list types we study, aggravating the problem. Moreover, LLAMA-3.1 has problems with negative numbers: here, the sign is often dropped when sorting, leading to a lack of faithfulness when sorting.

### 3.5 DeepSeek-r1 has problems with overthinking

Our expectation of strong performance for test-time reasoning models was not fulfilled by the DeepSeek-r1 model. The problem can be easily summarized: through the reasoning the model got distracted. The initial reasoning steps were elaborating on the task (e.g., "It seems like the input is a list of strings that should be sorted lexicographically"). When the reasoning stopped after this, the model performed the task as intended. Instead, the reasoning often continued for a long time, elaborating about unimportant details: in other words, the model was overthinking. While we have not checked all 1760 reasoning processes, we did a rough, qualitative analysis while studying parsing errors. Here, we observed three reasoning patterns that frequently emerged, that lead to different errors.

The first pattern was *overthinking how sorting works*. For example, for strings the model looked at the individual letters of the first two list items, comparing each with the other, then for the next list item, etc. This is similar to the misbehavior we also observed in our smaller reasoning analysis with o3-mini. This extended the context to such a degree, that afterwards the general instruction (sorting of the list as Python) was lost, and the model output something generic that describes how sorting works with the words from the start of the list as example. An example for this pattern is presented in Appendix A.6.

The second pattern was *looking for relationships between items*. This typically happened for numbers, where instead of just sorting the list, DeepSeek-r1 started to hallucinate and tried to find an equation that describes the list as a sequence. As a result, the output was not a list anymore, but rather some description of the pattern or something that should describe the relationship. Another example was that the model converted the strings of the NumberWords into actual numbers (breaking the list format) or instead of sorting the NumberWords lexicographically, they were instead sorted by the numeric values they represent. Notably, this even happened when the reasoning process initially correctly stated that the list is with strings and should be sorted lexicographically according to the task. This is another aspect we also observed with o3-mini.

The third pattern was the *list repetition*, that we also worried about initially: the reasoning process repeated the lists many times, and tried to check the sorting error in between. Through this, the result did not get better, but rather worse over time: with every iteration of the reasoning process, this added a risk of losing items or breaking the format, likelihood of a valid and faithful output. This pattern was not observed for

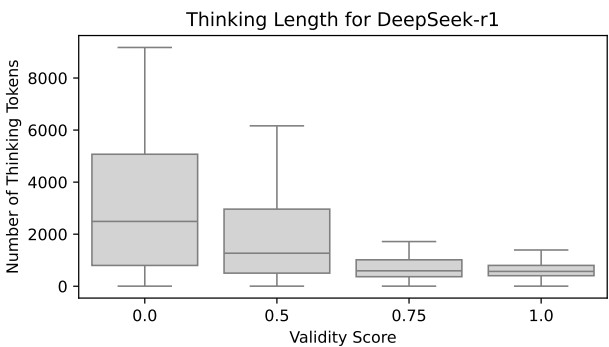

Figure 1: Boxplot of the number of tokens during the reasoning process for the different values of the $ValidityScore$.

o3-mini, which only repeated short subsequences of lists during the reasoning – though this may still have happened, and we just did not see this in the summaries that we had access to.

For all these patterns, it holds that the model generates more reasoning tokens than necessary for the task. This pushes the initial instruction from our prompt further away from the output, which often leads to hallucinations what the actual task is, resulting in those hallucinations that we described in the second pattern or instead just describing how sorting works. The data presented in Figure 1 confirms this: the mean number of reasoning tokens is by far the longest for outputs we could not parse, followed by cases that we could only parse with corner case handling. The cases that are almost valid or completely valid Python lists have by far the shortest reasoning processes.

The above observations explain the overall weak performance of DeepSeek-r1, which is driven by issues with the validity and faithfulness of the output, which just got lost in the reasoning process. However, if the output was indeed a list, the sorting was good (though still worse than for the large, proprietary models).

### 3.6 Long sequences are a problem

Across all tasks and models, the $SortBenchScore$ goes down with longer sequences. Moreover, when we consider the statistical analysis of the results for different list length (see Appendix A.2), we find that the differences between LLMs are insignificant for lists with at most 8 items. With 16 items, we start to observe a gap between the three best models (o3-mini, GPT-4o, and Claude-3.5-Sonnet) and the others. At this length, the overthinking problems of DeepSeek-r1 also start to manifest. At a length of 128, we still cannot measure a significant difference between o3-mini and GPT-4o, though Claude-3.5-Sonnet is different with a large effect size from the others. Only at length 256 all differences between the models become significant, except for Claude-3.5-Sonnet and Claude-3.5-Haiku. While the lack of a difference at this length between the two Claude variants may seem surprising, this is a consequence of the relatively small difference in sorting capability and the similar problems with producing valid outputs for list of this lengths that both models share.

When we analyze these trends over the $SortingScore$, $FaithfulnessScore$, and $ValidityScore$, we see that this mostly holds: having more list items increase the risk of invalid outputs, lead to the LLM being less faithful to the original list, and reduce the quality of the sorting. We say mostly, because some models actually show an increase in the $SortingScore$ for longer lists, e.g., DeepSeek-r1 for the advanced tasks. However, this does not really mean that the models get better at sorting with longer list. A careful look at the data reveals that this increase is accompanied by a sharp drop in the $FaithfulnessScore$, i.e., the LLMs just output some sorted output, instead of solving the actual task in these cases. Overall, this demonstrates quite clearly that long contexts are still a problem for current LLMs, while all LLMs are fairly good at dealing with short contexts.

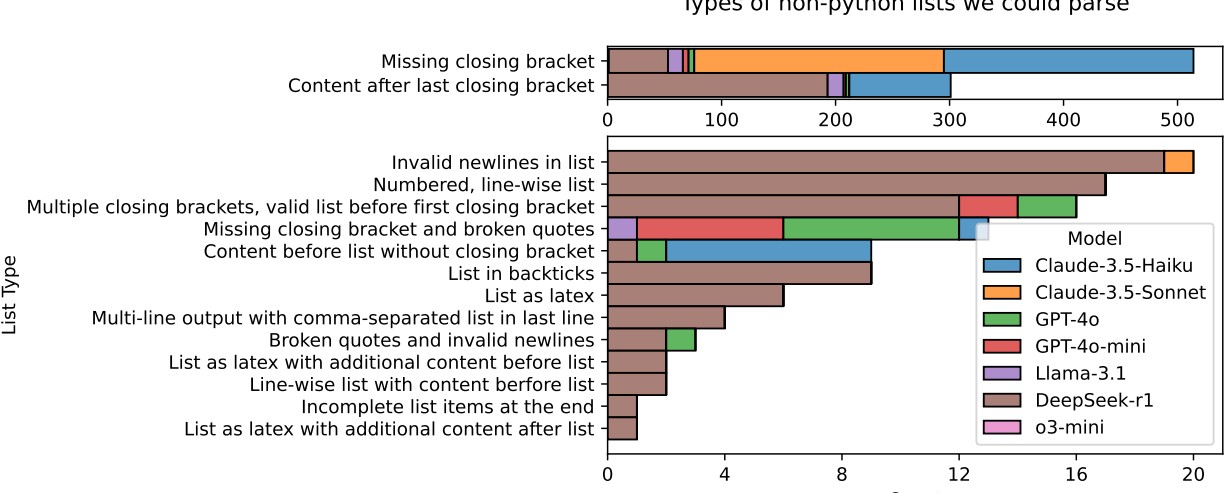

Figure 2: Different types of lists we found in the output of the LLMs that were not valid Python. The plot is split, since the first two types appear hundreds of times, while the others are rather corner cases with at most twenty instances.

A recent study by Modarressi et al. (2025) that specifically focused on such long-sequence tasks also found similar issues, including somewhat similar trends: the GPT-4o model was also most the stable with respect to longer sequences, all others dropped earlier. This indicates that achieving stronger performance at long contexts is possible as is demonstrated by OpenAI's models. Whether this is achieved by investing more compute time for training with longer contexts, tricks for achieving this during training without spending more compute, or some details in the model architecture, we cannot answer due to the lack of openness of OpenAI models. We note that the required compute budget is relatively small in comparison to the work by Modarressi et al. (2025), which indicates that our approach can be used as a relatively cheap option to explore performance improvements of models for long-context tasks.

### 3.7 Reasons for parsing errors

When we introduced the $ValidityScore$, we already stated that we observed many types of sorted lists, that were not the required Python lists. Figure 2 shows the errors we observed. In total, we defined fifteen customized list parsers to handle lists that were not valid Python. The most common problem were missing closing brackets, with about 200 instances for both Claude variants, and, additionally, the other models where this rather happened as a corner case. We also observed a variant of this where a missing closing bracket was combined with the quotes of a string list being broken (e.g., because there was no closing quote for the last list item). The other case that was driving most parsing errors is that there was content after the list. Such content was always an explanation for the sorting or the task that was performed. This happened very often with DeepSeek-r1, but was also not uncommon with Claude-3.5-Haiku. In general, content before or after the list was very often the problem with DeepSeek-r1, often also combined with other list types, including line-wise list (one item per line), enumerations (one item per line starting with the list index), lists in latex formats (e.g., \boxed or in math-mode), and in backticks to indicate code in MarkDown. Overall, "creatively" is the best description for how DeepSeek-r1 determined the output format.

### 3.8 Debug tasks show that some problems are persistent

The role of the first debug task, i.e., providing already sorted lists, is to gain insights into the question if problems with sorting are due to the difficulty of sorting task, or if simply following copying an already

sorted lists already leads to problems (see Appendix A.1 for detailed results). This also helps us to diagnose if the length of the context itself becomes a problem. The results indicate that the context length is not a problem and that following this instruction in general is not a problem: all models are able to provide fully sorted integer lists for all lengths. This is also mostly the case, though the Llama-3.1 model shows the same problem with longer sequences for floating point numbers (and a bit weaker) with English words we discussed above. Similary, DeepSeek has problems with the sorted English task, because it starts to overthink in exactly the same manner as for unsorted lists as input. We also observe the same problems with validity we have for unsorted lists. Overall, these debug tasks confirm that the problems we found with sorting a prevalent even in a simpler context and just further exacerbated by more difficult sorting tasks.

The second debug task, i.e., providing each list item twice, helps us to understand if such duplication negatively affects how the LLMs work. This is indeed the case. Even for the simple Int-0:1000 task, the sorting performance and faithfulness drops for most models. The exception are the very large model models (GPT-4o, o3-mini, and Claude-3.5-Sonnet). However, even these models have problems with the faithfulness for the English words, Claude-3.5 for Floating point numbers. This gives us a good indication about what is required to allow models to disambiguate two equal items: are very large number of parameters, though even this cannot fully solve this simple adversarial problem.

## 4 Discussion

### 4.1 SortBench Basic, Advanced, and Debug help us to understand LLMs

Our results show that SortBench is a suitable way to understand core properties of LLMs: Are they able to follow simple instructions, even if the output that is generated is long? Do the embeddings encode basic properties of tokens like their order in relation to items of the same type? In summary, our results show that test-time reasoning improves these capabilities (as expected), but comes with a risk of overthinking which may decreases the performance. Here, we observe a very large difference between the two test-time reasoning models in our benchmark: o3-mini has the overthinking mostly under control, while it is rampant for DeepSeek-r1. This shows both the potential, but also the risk of this sort of LLM training. Further, we showed that all LLMs, regardless of their size, can in principle sort lists (i.e., have this information in their embeddings), but that size gets important for longer lists. Interestingly, this means that even such a basic property (order) is not captured reliably over longer contexts, without huge models. Finally, this simple task demonstrates that you cannot trust LLMs to follow instructions. Even the (supposedly) simple debugging task of "sorting" an already sorted lists is typically not solved perfectly, due to the inherently random nature of LLMs.

### 4.2 Towards SortBench Hard

The sorting tasks we have considered here are still rather simple, yet sufficient to bring even current state-of-the-art models to their limits. However, o3-mini is close to solving this benchmark. Still, sorting offers many more options that can be used to test models, including multi-modal models that include vision. There are several such tasks that we foresee for a future hard version of the SortBench:

- Consideration of sorting orders (i.e. ascending and descending).

- Other ways to express numbers, e.g., as fractions or in the scientific exponential notation.

- Sorting within other languages than English.

- Sorting of long strings (e.g., sentences).

- Sorting with non-ASCII character sets.

- Sorting of well-known categories, such as cloth sizes (XS, S, M).

- Sorting of manually defined categories, i.e., the sorting order is defined as part of a prompt.

- Sorting of complex data types with multiple attributes for sorting (e.g., sorting two-column tabular data, first by one column, in case of ties by the second column.

- Sorting complex data with different sorting orders for different attributes.

- Sorting of items in images by their size (absolute in pixels).

- Sorting of items in images relative to a reference item (e.g., size in relation to a banana displayed in the image).

- Sorting of strings by the length of the strings.

- Sorting strings by their edit-distance to a baseline.

- Sorting items by their distance in an ontology.

As can be seen, this problem can easily be extended to require more complex skills, e.g., being able to apply multiple sorting orders at the same time, understanding manually defined sorting orders, and being able to develop suitable comparison mechanisms for complex relationships.

### 4.3 Avoiding overfitting with SortBench

An advantage of a benchmark like SortBench is that new data can be generated easily. This means that regular releases of this benchmark with new data, that was not yet seen by the models beforehand, are possible. We plan yearly releases of SortBench to prevent overfitting, which can easily occur if this data becomes part of the training of models. This is in stark contrast to commonly used benchmarks like MMLU (Hendrycks et al., 2020) which require a large amount of effort to create, update, and replace or the Chatbot Arena (Chiang et al., 2024) that requires users to constantly test and compare models.

### 4.4 Limitations

While SortBench provides good results and seems like a simple test that is yet capable of testing LLMs and bringing them to their limits, one may criticize whether the benchmark is actually useful, as it seemingly does not test any direct real-world use case, since no one would actually start sorting items this way. However, especially users without deep knowledge about how LLMs work or the domain in which they are using them, are often not aware of LLMs limitations. For example, users may use them to analyze tabular data, e.g., spreadsheets (Chen et al., 2024b). For such tasks, a basic understanding of orders, as well as the capability to faithfully handle long context-information is crucial. Further, one may argue that the scope is to narrow, since no other algorithmic tasks are used, e.g., string matching (Knuth et al., 1977). However, while the task we study is only sorting, the model capabilities we study also include the ability to be faithful to the task description, as well as understanding how the capabilities change for inputs of different lengths. While such considerations are certainly also possible with more tasks, the need to tailor the metrics and corner case analysis for parsing outputs (see Section 3.7) requires additional effort so that such an extension is non-trivial. Moreover, sorting also reveals interesting information about what is actually contained in the embeddings, i.e., whether the embeddings encode information like lexical or numerical order reliably. Finally, SortBench is computationally relatively cheap and can be scaled to any context length, while yielding valuable conclusions regarding the performance at different lengths in line with more complex benchmarks (Modarressi et al., 2025). Thus, we believe that SortBench is a suitable benchmark for LLMs that is easy to run, scale, and make more difficult for future LLM generations.

## 5 Conclusion

Sorting is a basic skill that humans learn at an early age. The SortBench benchmark tests the capabilities of LLMs at this task in different scenarios. We find that even current test-time reasoning models struggle at this task for longer lists, e.g., of random strings or very small floating point numbers. Moreover, test-time reasoning models tend to overthink and, e.g., ignore lexicographic orders if the strings represents numbers.

While this was mostly under control for o3-mini, the DeepSeek-r1 model we tested had big problems with this. This suggest that users and developers of test-time reasoning should carefully explore whether techniques to limit overthinking like those proposed by Chen et al. (2024a) control this for their tasks. SortBench further shows that all current models have problems with not changing the longer lists that should be sorted, an aspect we refer to as faithfulness. Overall, SortBench is a suitable benchmark to rank LLMs by their capability to follow instructions, handle inputs of different lengths, and solve a task that requires basic reasoning capabilities. The concept of automated data generation, result checking, and different complexity of inputs, and in-depth metrics for specific aspects of the results can also be extended to other algorithmic tasks, e.g., pattern matching.

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

# A  Appendix

This appendix provides additional details regarding our analysis with visualizations in Appendix A.1 and statistical analysis in Appendix A.2. Appendix A.3 provides additional details regarding the experiment setup with additional information regarding the impact of the temperature reported in Appendix A.4 and examples for each task in Appendix A.5. Finally, Appendix A.6 and Appendix A.7 presents one examples of overthinking of the test-time-reasoning models..

## A.1  Visualization of results

Within this section, we provide a visual analysis of the trends for all tasks and scores. Figures 3-10 show the results for the basic tasks, figures 7-10 for the advanced tasks, and figures 11-14 for the debug tasks. Each figure shows the trend line over the averages of the ten lists of the same length for each task.

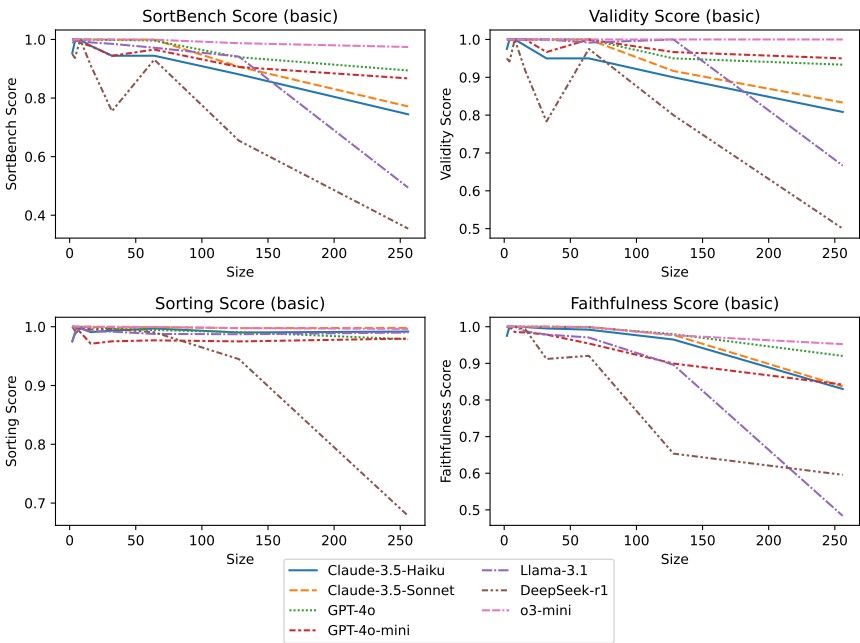

Figure 3: Aggregated results for all basic tasks by list size

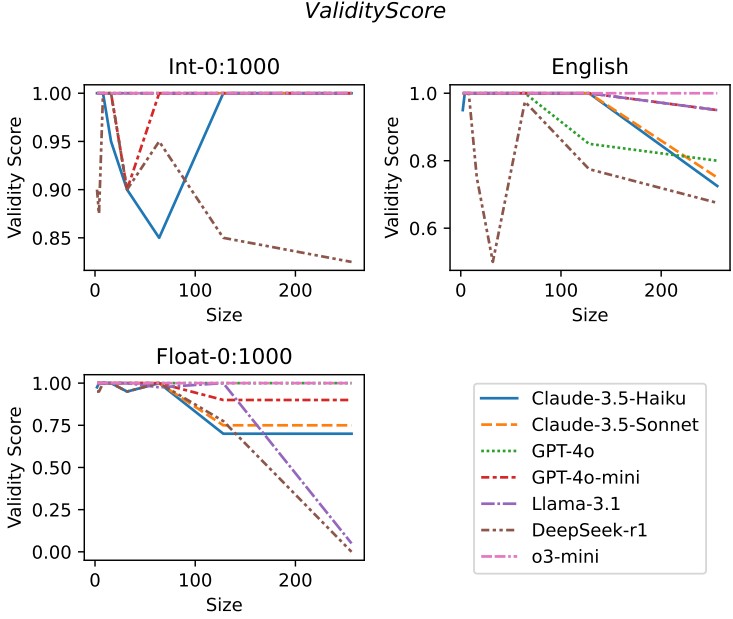

Figure 4: *ValidityScore* for all basic tasks by list size

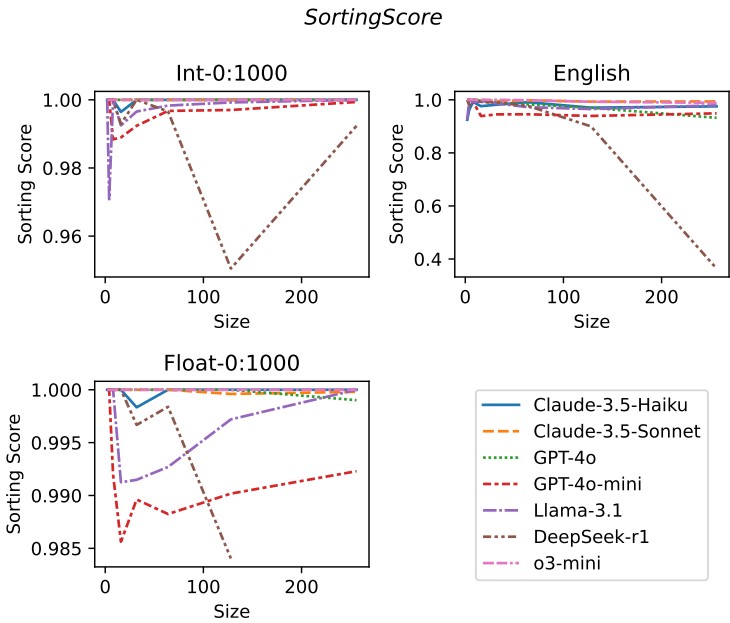

Figure 5: *SortingScore* for all basic tasks by list size

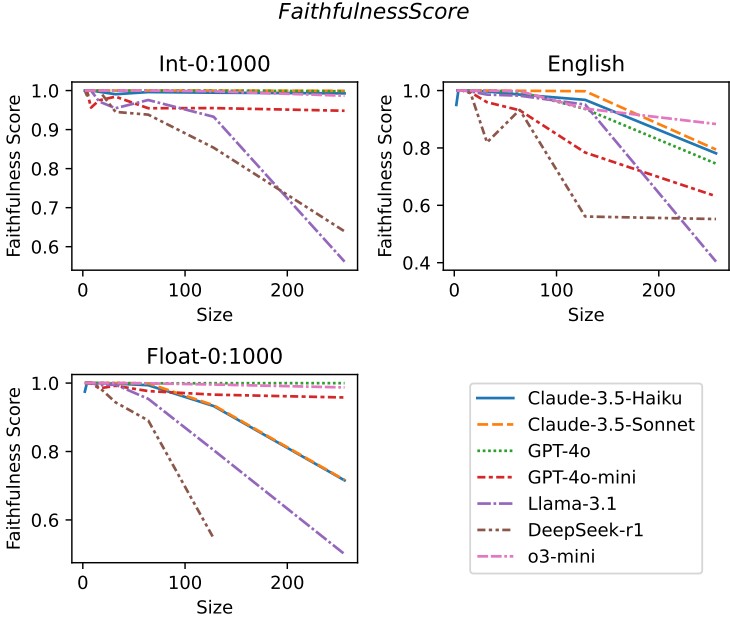

Figure 6: *FaithfulnessScore* for all basic tasks by list size

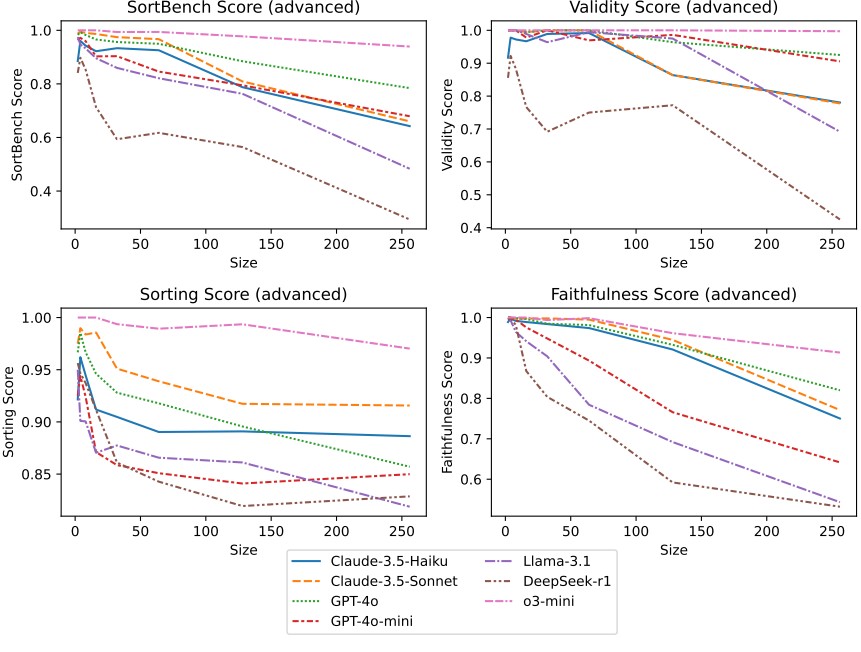

Figure 7: Aggregated results for all advanced tasks by list size

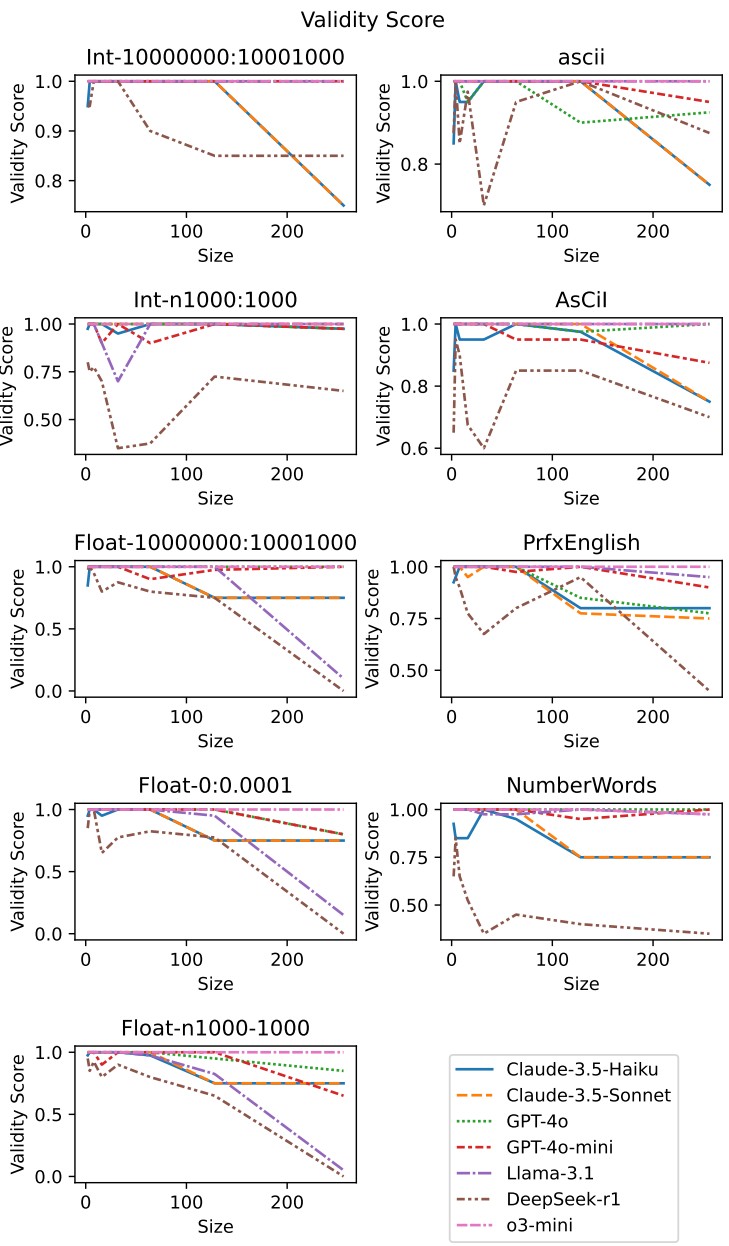

Figure 8: *ValidityScore* for all advanced tasks by list size

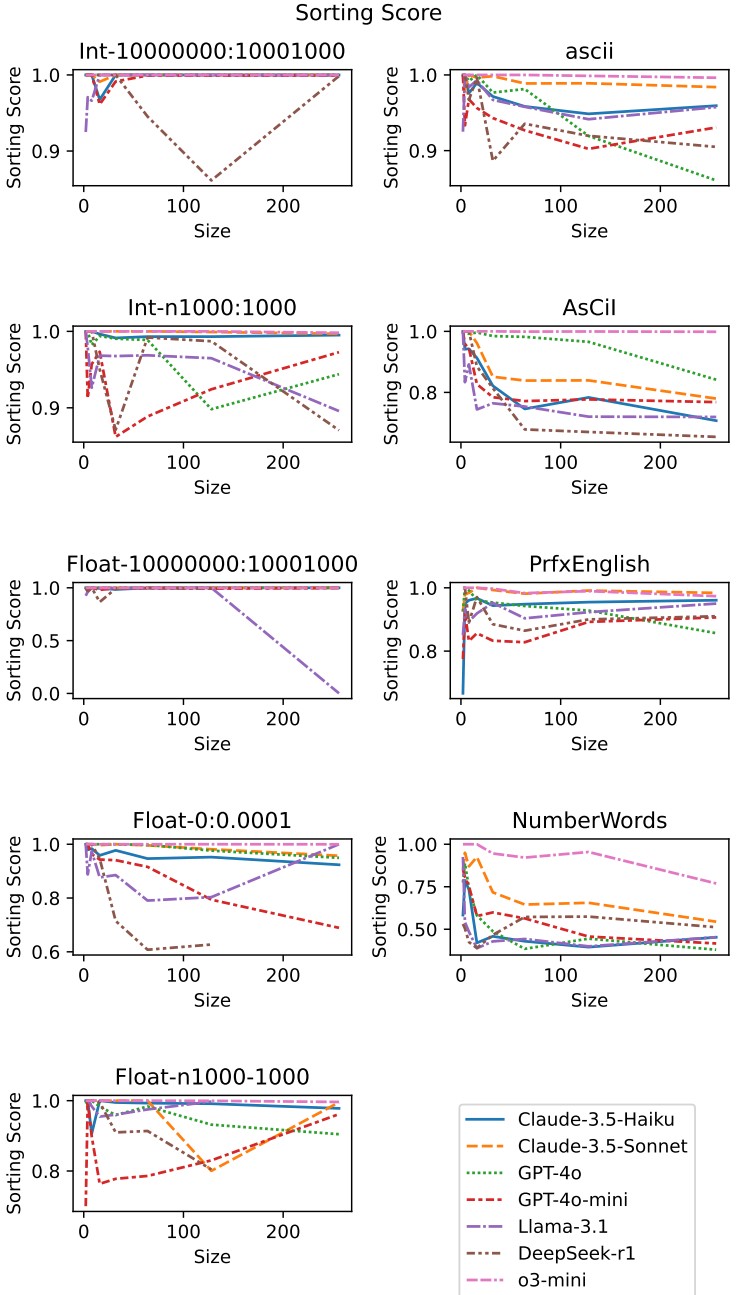

Figure 9: *SortingScore* for all advanced tasks by list size

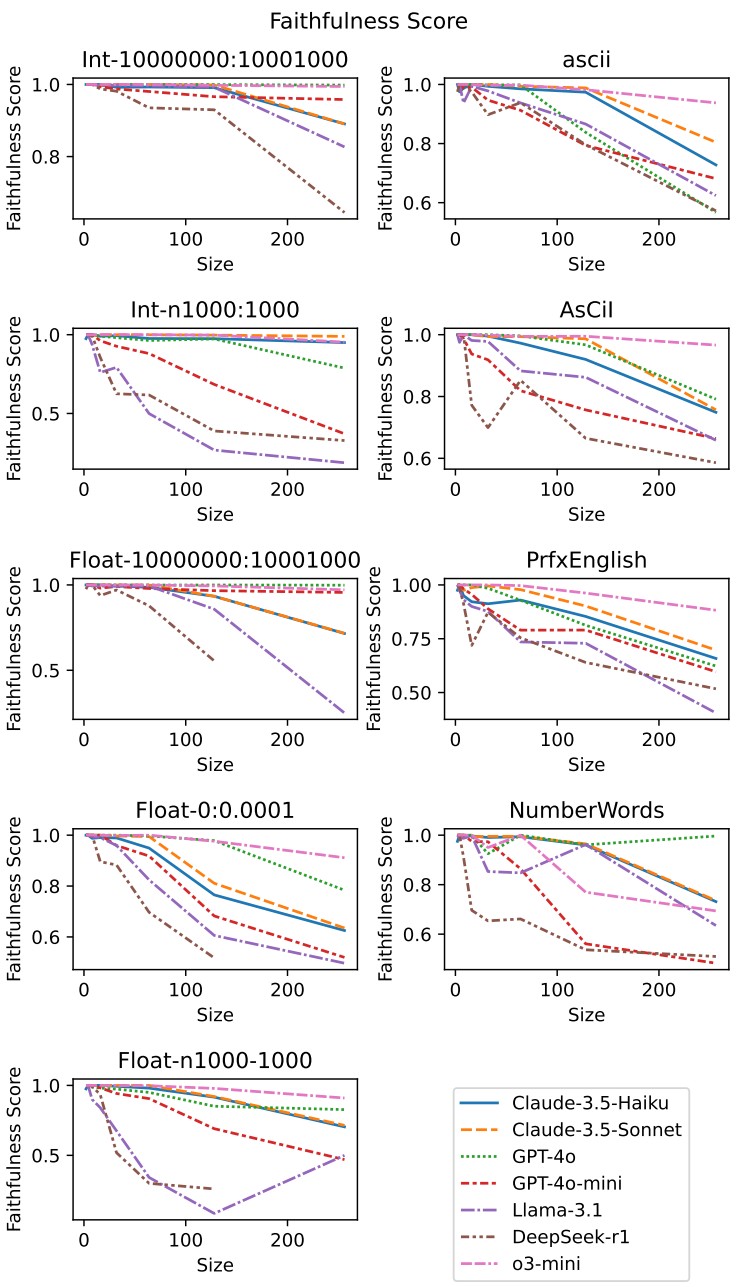

Figure 10: *FaithfulnessScore* for all advanced tasks by list size

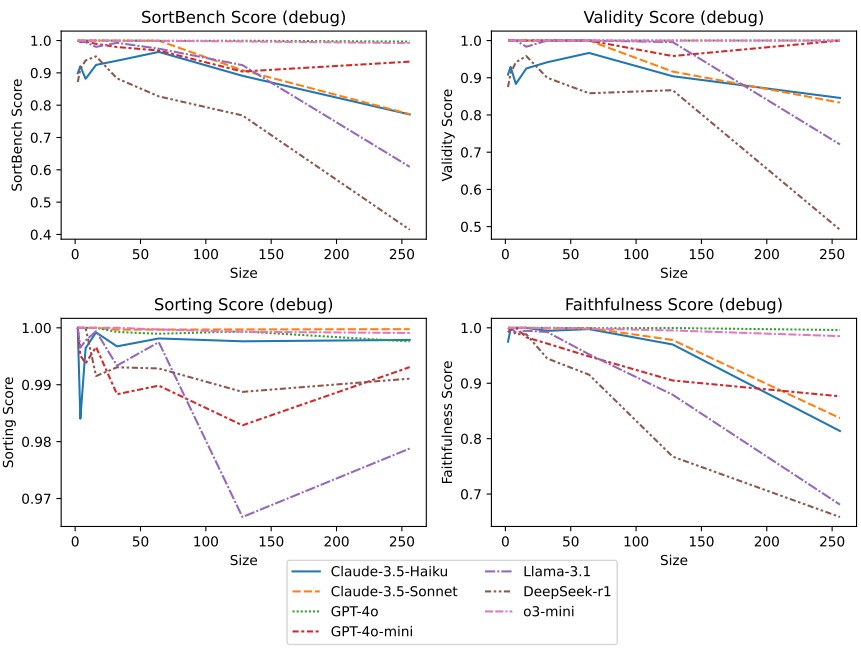

Figure 11: Aggregated results for all debug tasks by list size

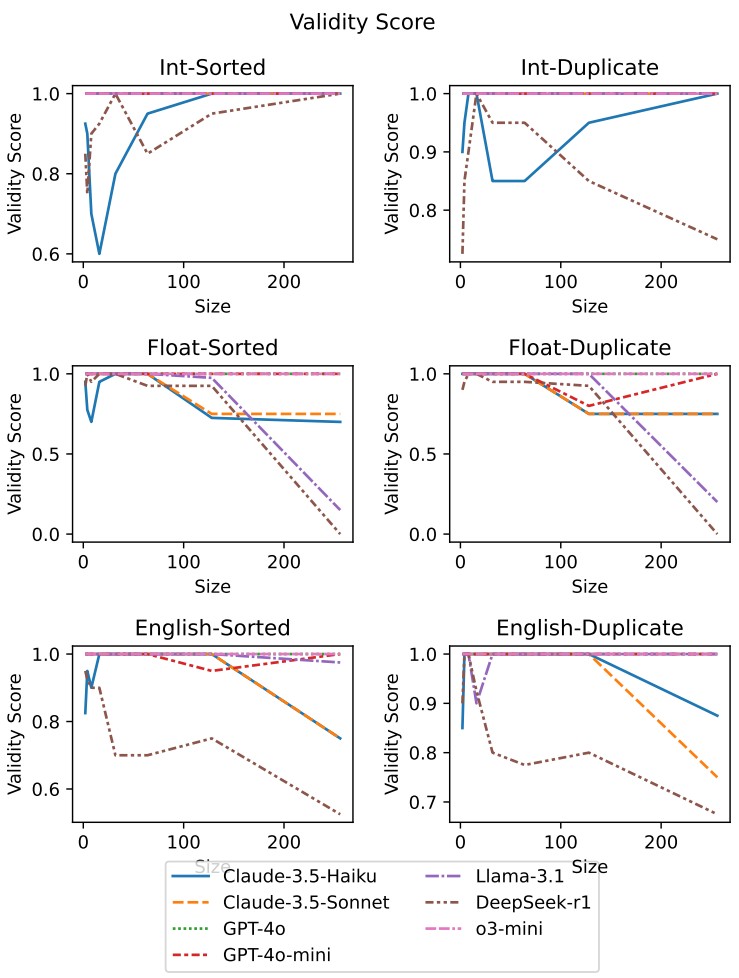

Figure 12: *ValidityScore* for all debug tasks by list size

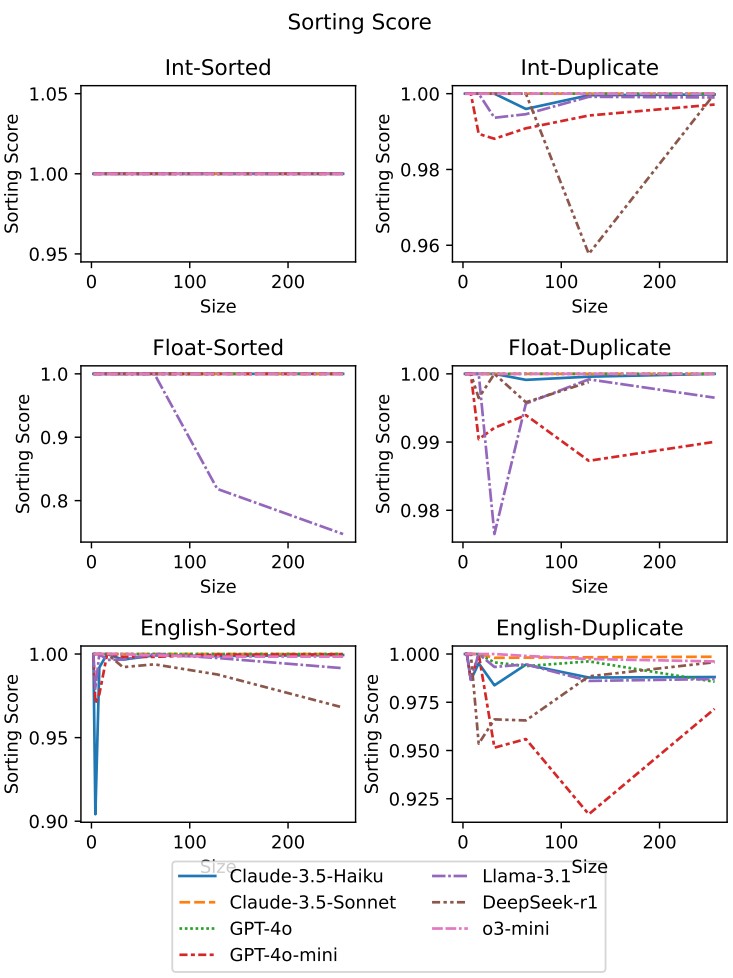

Figure 13: *SortingScore* for all debug tasks by list size

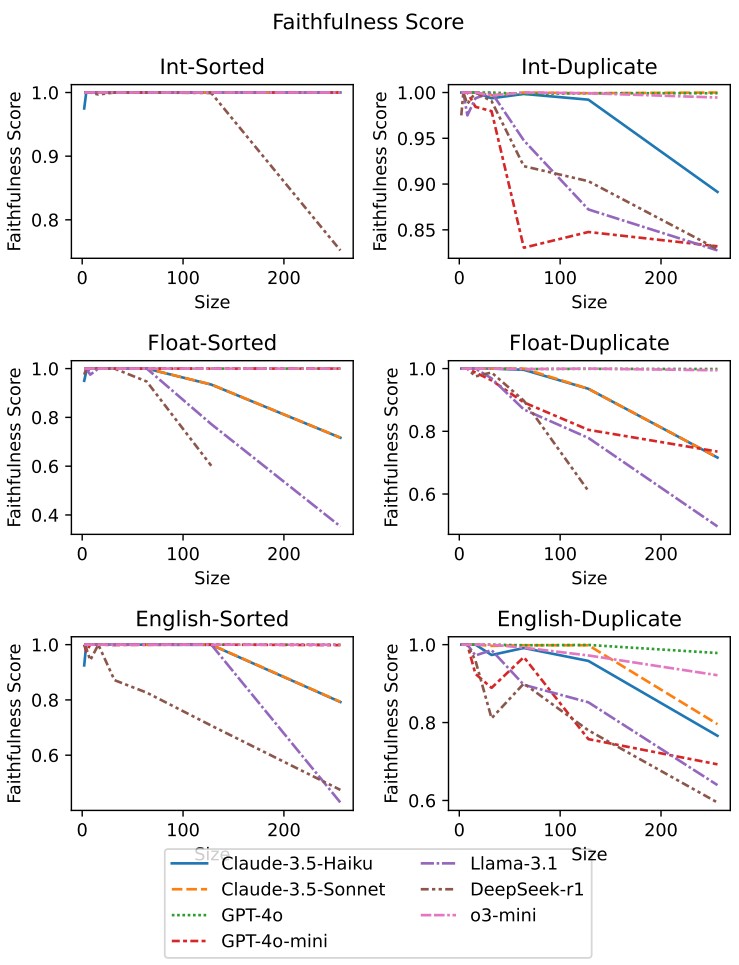

Figure 14: $Faithfulness Score$ for all debug tasks by list size

## A.2 Statistical analysis

This section presents the results of the statistical analysis of the *SortBenchScore* over all tasks for different list length. The Tables 2–9 report the mean value (M), standard deviation (SD), the 95% confidence interval (CI) of the mean value, Cohen's $d$ as effect size for the difference of means (Cohen, 2013) with respect to the best ranked model, and the probability that a model is smaller than the best ranked model ($p_{smaller}^{best}$) and the one reported in the line above ($p_{smaller}^{above}$), computed with a Bayesian signed rank test Benavoli et al. (2014) with a Region of Practical Equivalence (ROPE) as $\pm 0.1 \cdot d$ following Kruschke & Liddell (2018). The effect size is only reported in relation to the best-ranked model, but only if there is at least a 80% probability, that a the mean performance is smaller than that of the best-ranked model. The computation of the confidence intervals uses Bonferroni-Dunn correction for the number of models to ensure that the family-wise error for each list length is 0.05.

This reporting follows the guidelines by Benavoli et al. (2017), and updated version of the popular guidelines by Demšar (2006).

| Model | M | SD | CI | $d$ | $p_{smaller}^{best}$ | $p_{smaller}^{above}$ |
|---|---|---|---|---|---|---|
| o3-mini | 1.000 | 0.000 | [1.000, 1.000] | - | - | |
| Claude-3.5-Sonnet | 0.994 | 0.048 | [0.982, 1.006] | - | 0.000 | 0.000 |
| GPT-4o | 0.992 | 0.055 | [0.978, 1.005] | - | 0.000 | 0.000 |
| Llama-3.1 | 0.985 | 0.073 | [0.967, 1.004] | - | 0.000 | 0.000 |
| GPT-4o-mini | 0.981 | 0.082 | [0.961, 1.002] | - | 0.000 | 0.000 |
| Claude-3.5-Haiku | 0.896 | 0.187 | [0.850, 0.943] | - | 0.007 | 0.003 |
| DeepSeek-r1 | 0.867 | 0.232 | [0.809, 0.925] | - | 0.073 | 0.029 |

Table 2: Statistics for lists of length 2 for all tasks

| Model | M | SD | CI | $d$ | $p_{smaller}^{best}$ | $p_{smaller}^{above}$ |
|---|---|---|---|---|---|---|
| o3-mini | 1.000 | 0.000 | [1.000, 1.000] | - | - | - |
| Claude-3.5-Sonnet | 0.997 | 0.017 | [0.993, 1.002] | - | 0.000 | 0.000 |
| GPT-4o | 0.996 | 0.026 | [0.990, 1.003] | - | 0.000 | 0.000 |
| GPT-4o-mini | 0.984 | 0.055 | [0.971, 0.998] | - | 0.000 | 0.000 |
| Llama-3.1 | 0.972 | 0.072 | [0.955, 0.990] | - | 0.000 | 0.000 |
| Claude-3.5-Haiku | 0.953 | 0.135 | [0.919, 0.986] | - | 0.000 | 0.000 |
| DeepSeek-r1 | 0.909 | 0.193 | [0.861, 0.957] | - | 0.009 | 0.004 |

Table 3: Statistics for lists of length 4 for all tasks

| Model | M | SD | CI | $d$ | $p_{smaller}^{best}$ | $p_{smaller}^{above}$ |
|---|---|---|---|---|---|---|
| o3-mini | 1.000 | 0.000 | [1.000, 1.000] | - | - | - |
| Claude-3.5-Sonnet | 0.995 | 0.026 | [0.989, 1.002] | - | 0.000 | 0.000 |
| GPT-4o | 0.991 | 0.039 | [0.982, 1.001] | - | 0.000 | 0.000 |
| GPT-4o-mini | 0.977 | 0.047 | [0.965, 0.989] | - | 0.083 | 0.203 |
| Llama-3.1 | 0.964 | 0.082 | [0.944, 0.984] | - | 0.102 | 0.089 |
| Claude-3.5-Haiku | 0.932 | 0.162 | [0.891, 0.972] | - | 0.005 | 0.033 |
| DeepSeek-r1 | 0.906 | 0.217 | [0.852, 0.960] | - | 0.003 | 0.000 |

Table 4: Statistics for lists of length 8 for all tasks

| Model | M | SD | CI | $d$ | $p_{smaller}^{best}$ | $p_{smaller}^{above}$ |
|---|---|---|---|---|---|---|
| o3-mini | 1.000 | 0.000 | [1.000, 1.000] | - | - | - |
| Claude-3.5-Sonnet | 0.993 | 0.041 | [0.983, 1.003] | - | 0.000 | 0.000 |
| GPT-4o | 0.983 | 0.068 | [0.966, 1.000] | - | 0.000 | 0.000 |
| GPT-4o-mini | 0.945 | 0.127 | [0.913, 0.976] | 0.616 | 1.000 | 0.996 |
| Llama-3.1 | 0.940 | 0.138 | [0.906, 0.974] | 0.615 | 0.816 | 0.021 |
| Claude-3.5-Haiku | 0.932 | 0.158 | [0.893, 0.972] | - | 0.042 | 0.001 |
| DeepSeek-r1 | 0.827 | 0.292 | [0.754, 0.900] | - | 0.620 | 0.477 |

Table 5: Statistics for lists of length 16 for all tasks

| Model | M | SD | CI | $d$ | $p_{smaller}^{best}$ | $p_{smaller}^{above}$ |
|---|---|---|---|---|---|---|
| o3-mini | 0.997 | 0.027 | [0.990, 1.003] | - | - | - |
| Claude-3.5-Sonnet | 0.987 | 0.044 | [0.976, 0.998] | - | 0.000 | 0.000 |
| GPT-4o | 0.978 | 0.074 | [0.959, 0.996] | - | 0.000 | 0.000 |
| Claude-3.5-Haiku | 0.937 | 0.139 | [0.902, 0.971] | - | 0.565 | 0.004 |
| GPT-4o-mini | 0.936 | 0.112 | [0.908, 0.964] | 0.750 | 1.000 | 0.965 |
| Llama-3.1 | 0.925 | 0.163 | [0.884, 0.966] | 0.613 | 1.000 | 0.018 |
| DeepSeek-r1 | 0.719 | 0.358 | [0.630, 0.808] | 1.097 | 1.000 | 1.000 |

Table 6: Statistics for lists of length 32 for all tasks

| Model | M | SD | CI | $d$ | $p_{smaller}^{best}$ | $p_{smaller}^{above}$ |
|---|---|---|---|---|---|---|
| o3-mini | 0.996 | 0.025 | [0.990, 1.003] | - | - | - |
| Claude-3.5-Sonnet | 0.983 | 0.051 | [0.971, 0.996] | - | 0.000 | 0.000 |
| GPT-4o | 0.974 | 0.073 | [0.956, 0.992] | - | 0.007 | 0.002 |
| Claude-3.5-Haiku | 0.942 | 0.124 | [0.911, 0.973] | 0.610 | 0.985 | 0.003 |
| GPT-4o-mini | 0.907 | 0.143 | [0.871, 0.943] | 0.872 | 1.000 | 1.000 |
| Llama-3.1 | 0.897 | 0.152 | [0.860, 0.935] | 0.907 | 1.000 | 0.406 |
| DeepSeek-r1 | 0.742 | 0.318 | [0.662, 0.822] | 1.129 | 1.000 | 0.999 |

Table 7: Statistics for lists of length 64 for all tasks

| Model | M | SD | CI | $d$ | $p_{smaller}^{best}$ | $p_{smaller}^{above}$ |
|---|---|---|---|---|---|---|
| o3-mini | 0.986 | 0.047 | [0.974, 0.998] | - | - | - |
| GPT-4o | 0.932 | 0.167 | [0.890, 0.974] | - | 0.001 | 0.001 |
| Claude-3.5-Sonnet | 0.859 | 0.157 | [0.819, 0.898] | 1.095 | 1.000 | 1.000 |
| GPT-4o-mini | 0.850 | 0.197 | [0.801, 0.899] | 0.950 | 1.000 | 0.578 |
| Llama-3.1 | 0.846 | 0.180 | [0.802, 0.891] | 1.057 | 1.000 | 0.809 |
| Claude-3.5-Haiku | 0.838 | 0.165 | [0.796, 0.879] | 1.217 | 1.000 | 0.885 |
| DeepSeek-r1 | 0.647 | 0.287 | [0.576, 0.719] | 1.646 | 1.000 | 1.000 |

Table 8: Statistics for lists of length 128 for all tasks

| Model | M | SD | CI | $d$ | $p_{smaller}^{best}$ | $p_{smaller}^{above}$ |
|---|---|---|---|---|---|---|
| o3-mini | 0.963 | 0.072 | [0.945, 0.981] | - | - | - |
| GPT-4o | 0.873 | 0.220 | [0.819, 0.928] | 0.546 | 0.956 | 0.956 |
| GPT-4o-mini | 0.796 | 0.240 | [0.737, 0.856] | 0.940 | 1.000 | 1.000 |
| Claude-3.5-Sonnet | 0.716 | 0.159 | [0.676, 0.756] | 1.998 | 1.000 | 1.000 |
| Claude-3.5-Haiku | 0.702 | 0.167 | [0.661, 0.744] | 2.027 | 1.000 | 0.000 |
| Llama-3.1 | 0.528 | 0.362 | [0.437, 0.619] | 1.664 | 1.000 | 1.000 |
| DeepSeek-r1 | 0.345 | 0.366 | [0.253, 0.436] | 2.345 | 1.000 | 1.000 |

Table 9: Statistics for lists of length 256 for all tasks

### A.3 Details on Experiment Setup

The models from OpenAI (o3-mini, GPT-4o, GPT-4o-mini) and Anthropic (Claude-3.5-Sonnet, Claude-3.5-Haiku) were used using the APIs. The open-weights models (Llama-3.1, DeepSeek-r1) were used on a local inference server that also exposed an OpenAI comptatible API with A100 accelerators. The system and user prompts (see Section 2.1) were configured using the respective chat completions APIs. All experiments were executed with the default temperature of 1, which mimics the most common and setting in which LLMs are used. We did not re-run the experiments, i.e., we did not explore the randonmness when sorting the same list multiple times with the same model. Instead, we explored the randomness with respect to sorting ten lists with the same properties (same task, same length). Exploring both at the same time was not feasible given our available compute budget.

### A.4 Impact of Temperature

We repeated our experiments with a temperature of 0 for the GPT-4o-mini model to understand the impact of this choice on the results. Table 10 reports the difference in results between a temperature of 0 (i.e. no randomness and always select the most likely token) and the default temperature of 1 (i.e., selecting tokens based on their probability as determined by the LLM without further scaling). As can be seen, the impact of the temperature on the results is very small, which is not surprising, because if the models can understand order should not be probabilistic property. Hence, the probabilities should be very close to one, which means the outcome is not strongly affected by lowering the temperature to 0. We note that the changes in performance are so small, that they would not lead to any rank changes for any of the tasks and the that magnitude of the changes is well below the confidence intervals reported for the different list lengths for all tasks in Appendix A.2.

| Tasks | Model | $ModelScore$ | $SortingScore$ | $FaithfulnessScore$ | $ValidityScore$ |
|---|---|---|---|---|---|
| Basic | Temperature 0 | 0.920 | 0.975 | 0.885 | 0.979 |
|  | Temperature 1 | 0.901 | 0.978 | 0.888 | 0.964 |
| Advanced | Temperature 0 | 0.760 | 0.849 | 0.718 | 0.955 |
|  | Temperature 1 | 0.758 | 0.851 | 0.744 | 0.945 |
| Debug | Temperature 0 | 0.942 | 0.987 | 0.899 | 0.997 |
|  | Temperature 1 | 0.938 | 0.990 | 0.905 | 0.990 |
| All tasks | Temperature 0 | 0.847 | 0.916 | 0.806 | 0.973 |
|  | Temperature 1 | 0.842 | 0.919 | 0.823 | 0.963 |

Table 10: Impact of temperature on the results of the GPT-4o-mini results.

### A.5 Examples for inputs for each taks

To give a better impression of the data for each task, we show the first list of length four for each task below.

- Int-0:1000: [587, 168, 137, 250]

- Float-0:1000:                [719.1599272297124, 226.56598806952388, 224.7509100064007, 419.19791780439385]

- English:                ['salafi_movement', 'muscle-bound', 'plagianthus_betulinus', 'family_palmae']

- Int-10000000:10001000: [10000772, 10000236, 10000122, 10000109]

- Float-10000000:10001000:  [10000017.48054105, 10000728.575737009, 10000155.596665977, 10000484.293734727]

- Float-0:0.0001:                           `[9.350145422496336e-05, 5.4316978103712525e-05, 4.082949301074651e-05, 9.662659205067486e-05]`

- Int-n1000:1000: `[-2758, -2810, 119, -4992]`

- Float-n1000:1000:        `[-9925.009120088629, -4225.007770637259, -4285.608502255741, -3562.1887107798702]`

- ascii: `['jlvfq', 'yldfh', 'yqhra', 'xwruy']`

- AsCiI: `['CKqaD', 'cCtwz', 'kUtxr', 'CvQLj']`

- PrfxEnglish: `['VVVspike_rush', 'VVVdieback', 'VVVcuria', 'VVVordinance']`

- NumberWords:   `['six hundred and thirty-six', 'four hundred and twenty-six', 'five hundred and sixty-three', 'seven hundred and sixty-six']`

## A.6 Example of overthinking by DeepSeek-r1

Below is an example of overthinking by DeepSeek-r1 for the fifth list of length 32 of the English task.

```
<think>
Okay, so I have this list of words that I need to sort. Let me look at them first. The list is: ['
    program', 'w._c._handy', 'greater_celandine', 'greco-roman_deity', 'vespid_wasp', 'fraternity',
    'enallage', 'personality_assessment', 'blowing_up', 'compline', 'carboniferous', 'surface_fire
    ', 'capsicum_annuum_longum', 'ft', 'ungathered', 'non-proliferation', 'cyprinid_fish', '
    pennatulidae', 'episome', 'cockeyed', 'amalgamator', 'pope', 'blue_columbine', 'slam_dancing',
    'highbrow', 'william_dean_howells', 'pukka', 'miasma', 'rhizomorph', 'ground_rule', 'hans_c._j.
    _gram', 'news_item'].

I need to sort them according to the common sorting used in Python. I remember that in Python, when
    you sort strings, it uses lexicographical order based on ASCII values by default. So, it's
    like dictionary order, comparing each character one by one.

First, I should check if there are any special characters or cases that might affect the sorting.
    Looking through the list, I notice some words have underscores, hyphens, periods, and capital
    letters in the middle. For example, 'w._c._handy' has periods and underscores, and 'greco-
    roman_deity' has a hyphen.

In ASCII, uppercase letters come before lowercase letters, so any word with an uppercase letter
    might come before others that start with lowercase. But wait, all these words are in lowercase
    except for the special characters. Wait no, actually, looking again, some have capital letters
    inside like 'william_dean_howells'no, it's all lowercase. So maybe case isn't an issue here.

Next, special characters: periods and underscores. I think in ASCII, the underscore comes after
    most symbols but before letters. Let me confirm the ASCII order to be sure.

Looking up ASCII table:
- The period '.' has a code of 46.
- The hyphen '-' is 45.
- The underscore '_' is 95.
- Lowercase letters start at 'a' which is 97, so all lowercase letters come after these symbols.

So when sorting, any word starting with a symbol will come before words starting with letters. But
    in this list, the only words starting with symbols are 'w._c._handy', which starts with 'w' but
     has periods and underscores later on, but not at the beginning.

Wait, no. The first character is 'w', so it's a letter. So that word will be compared based on 'w'.
    Let me see: words starting with letters from a to z.
```

I need to sort them alphabetically. I can try to list them in order.

Let me write down the words and see their starting letters:

- amalgamator (a)
- blue_columbine (b)
- blowing_up (b)
- capsicum_annuum_longum (c)
- carboniferous (c)
- cockeyed (c)
- compline (c)
- cyprinid_fish (c)
- enallage (e)
- episome (e)
- fraternity (f)
- ft (f)
- greco-roman_deity (g)
- greater_celandine (g)
- ground_rule (g)
- hans_c._j._gram (h)
- highbrow (h)
- miasma (m)
- news_item (n)
- non-proliferation (n)
- pennatulidae (p)
- personality_assessment (p)
- pope (p)
- program (p)
- pukka (p)
- rhizomorph (r)
- slam_dancing (s)
- surface_fire (s)
- ungathered (u)
- vespid_wasp (v)
- w._c._handy (w)

Wait, but I might have missed some. Let me go through the original list again to make sure.

Original list:

'program', 'w._c._handy', 'greater_celandine', 'greco-roman_deity', 'vespid_wasp', 'fraternity', 'enallage', 'personality_assessment', 'blowing_up', 'compline', 'carboniferous', 'surface_fire', 'capsicum_annuum_longum', 'ft', 'ungathered', 'non-proliferation', 'cyprinid_fish', 'pennatulidae', 'episome', 'cockeyed', 'amalgamator', 'pope', 'blue_columbine', 'slam_dancing', 'highbrow', 'william_dean_howells', 'pukka', 'miasma', 'rhizomorph', 'ground_rule', 'hans_c._j._gram', 'news_item'

Wait, I missed some words in my initial grouping. Let me list them all with their starting letters:

1. amalgamator (a)
2. blue_columbine (b)
3. blowing_up (b)
4. capsicum_annuum_longum (c)
5. carboniferous (c)
6. cockeyed (c)
7. compline (c)
8. cyprinid_fish (c)
9. enallage (e)

```
10. episome (e)
11. fraternity (f)
12. ft (f)
13. greco-roman_deity (g)
14. greater_celandine (g)
15. ground_rule (g)
16. hans_c._j._gram (h)
17. highbrow (h)
18. miasma (m)
19. news_item (n)
20. non-proliferation (n)
21. pennatulidae (p)
22. personality_assessment (p)
23. pope (p)
24. program (p)
25. pukka (p)
26. rhizomorph (r)
27. slam_dancing (s)
28. surface_fire (s)
29. ungathered (u)
30. vespid_wasp (v)
31. w._c._handy (w)
```

Wait, that's 31 items? Let me count: from 'program' as the first to 'news_item' at the end. Yeah, I think so.

Now, I need to sort them in alphabetical order based on their starting letters.

Starting with a: amalgamator is first.

Then b: blue_columbine and blowing_up. Which comes first? 'blue' vs 'blowing'. 'blue' (b-l-u-e) comes before 'blow' (b-l-o-w). Because after 'blu', 'e' comes before 'o'.

So order would be blue_columbine, then blowing_up.

Next c: several words. Let's list them:

capsicum_annuum_longum, carboniferous, cockeyed, compline, cyprinid_fish.

Now, within the c group, we need to sort each word by their next letters after 'c'.

capsicum... (c-a) comes before carboniferous (c-a-r), but wait:

Wait, capsicum starts with 'cap', carboniferous with 'car'. So 'a' vs 'r': 'ca' comes before 'car'? No, because 'a' is earlier in the alphabet than 'r'.

Wait no, 'capsicum' is c-a-p-s..., while 'carboniferous' is c-a-r-b... Wait, after 'c', both have 'a' as the second letter. Then third letters: 'p' vs 'r'. So 'p' comes before 'r', so capsicum comes before carboniferous.

But wait, let me check:

capsicum_annuum_longum starts with 'cap'

carboniferous starts with 'car'

So yes, 'cap' comes before 'car'.

Then next is cockeyed (c-o-c-k-e-y-e-d). Wait no: after 'c', the second letter in cockeyed is 'o',
    which is higher than 'a' and 'r'. So wait, let me clarify.

Wait, I think I made a mistake. The c group includes:

capsicum_annuum_longum (starts with c-a)
carboniferous (c-a-r)
cockeyed (c-o)
compline (c-o-m)
cyprinid_fish (c-y)

So the order within 'c' would be:

1. capsicum_annuum_longum
2. carboniferous
3. cockeyed
4. compline
5. cyprinid_fish

Because after 'c', 'a' comes before 'o' and 'y'. So first, all 'ca' words.

Between capsicum and carboniferous: both start with 'ca'.

So next letters:

capsicum: c-a-p-s...

carboniferous: c-a-r-b...

So third letter: p vs r. Since p (16th) comes before r (18th), so capsicum comes first, then
    carboniferous.

Then the next words in c group are cockeyed and compline, both starting with 'co'.

Compare cockeyed (c-o-c-k...) and compline (c-o-m-p...).

After 'c-o', third letters: 'c' vs 'm'. 'c' comes before 'm', so cockeyed comes before compline.

Then cyprinid_fish starts with c-y, which is after all the others in 'c' group.

So within 'c' group:

1. capsicum_annuum_longum

2. carboniferous

3. cockeyed

4. compline

5. cyprinid_fish

Next is e: enallage and episome.

enallage (e-n) comes before episome (e-p).

Then f: fraternity and ft.

fraternity starts with 'fra', which comes before 'ft' because after 'f', 'r' comes before 't'.

So fraternity, then ft.

Next g: greco-roman, greater, ground.

Wait, the words are:

greco-roman_deity? Wait no, original word is 'greco-roman_deity'? No, looking back at list:

No, from the user's message, the third word is 'greater_celandine', fourth is 'greco-roman_deity'.

Wait, actually in the initial list provided by the user:

The words are: program, w._c._handy, greater_celandine, greco-roman_deity, vespid_wasp, fraternity, enallage, personality_assessment, blowing_up, compline, carboniferous, surface_fire, capsicum_annuum_longum, ft, ungathered, non-proliferation, cyprinid_fish, pennatulidae, episome , cockeyed, amalgamator, pope, blue_columbine, slam_dancing, highbrow, william_dean_howells, pukka, miasma, rhizomorph, ground_rule, hans_c._j._gram, news_item.

Wait I think I misassigned some words earlier. Let me correct:

From the user's original list, the order is as provided. So when grouping by first letter, we have to consider all the words accurately.

But perhaps it's easier to just go through each word and sort them one by one.

Let me make a list of all 31 words with their starting letters:

1. program (p)
2. w._c._handy (w)
3. greater_celandine (g)
4. greco-roman_deity (g)
5. vespid_wasp (v)
6. fraternity (f)
7. enallage (e)
8. personality_assessment (p)
9. blowing_up (b)
10. compline (c)
11. carboniferous (c)
12. surface_fire (s)
13. capsicum_annuum_longum (c)
14. ft (f)
15. ungathered (u)
16. non-proliferation (n)
17. cyprinid_fish (c)
18. pennatulidae (p)
19. episome (e)
20. cockeyed (c)
21. amalgamator (a)
22. pope (p)
23. blue_columbine (b)
24. slam_dancing (s)
25. highbrow (h)
26. william_dean_howells (w) [Wait, original word is 'william_dean_howells'? Or perhaps it's a typo and should be 'howell' or similar? But regardless, starts with 'w'.]
27. pukka (p)
28. miasma (m)
29. rhizomorph (r)
30. ground_rule (g)

```
31. hans_c._j._gram (h)
32. news_item (n)
```

Wait, now I count 32 items? Hmm, perhaps the user had a typo in the original list. Alternatively, perhaps I miscounted.

But regardless, let's proceed with each word and assign their starting letters correctly.

Now, to sort them all alphabetically:

The order would be based on first letter, then second, etc.

So starting from 'a', which is only one word: amalgamator.

Then 'b': blowing_up and blue_columbine. Between these two, 'blowing' comes before 'blue' because after 'bl', the next letters are 'o' vs 'u'. So blowing_up (9) comes before blue_columbine (23).

Next is 'c' group: compline (10), carboniferous (11), capsicum_annuum_longum (13), cyprinid_fish (17), cockeyed (20). Let's sort these:

- The first letters are all 'c', so look at second letter.

compline starts with 'co'

carboniferous: 'ca'

capsicum: 'ca'

cyprinid: 'cy'

cockeyed: 'co'

So arrange them by the second letter. Letters in order: a, o, y.

So first, carboniferous and capsicum (both start with 'ca'), then compline and cockeyed ('co'), then cyprinid_fish.

Between carboniferous and capsicum:

carboniferous vs capsicum. Both start with 'c', 'a'. Third letters: 'r' in carboniferous, 'p' in capsicum. So 'cap' comes before 'car' because p (16) < r (18). So capsicum comes first.

So within 'ca' group:

capsicum_annuum_longum (13)

carboniferous (11)

Then the 'co' group: compline vs cockeyed.

compline starts with 'com', cockeyed with 'coc'. So after 'c-o', third letter: 'm' in compline, 'c' in cockeyed. Wait no:

Wait, compline is spelled c-o-m-p-l-i-n-e

cockeyed is c-o-c-k-e-y-e-d.

So the first letters are both 'co'.

The third letters: m vs c? No, wait: let's compare letter by letter beyond the second letter.

compline: c o m p l i n e

cockeyed: c o c k e y e d

After 'c o', next is 'm' vs 'c'. Since 'c' comes before 'm', cockeyed (20) would come after
    carboniferous and capsicum, but wait no. Wait the initial sort is by first letters, then second
     letters, etc.

Wait in alphabetical order, it's:

After 'ca' words, we have 'co', which starts with 'c o'. So all 'co' words come after 'ca'.

Within 'co', compare each letter beyond that.

compline: c o m p l i n e

cockeyed: c o c k e y e d

So the third letters are m and c. Since c comes before m, cockeyed (c o c ...) would come before
    compline (c o m ...). Therefore:

capsicum_annuum_longum (13)

carboniferous (11)

cockeyed (20)

compline (10)

Then cyprinid_fish (17) as it starts with 'cy'.

So the 'c' group sorted is: 13,11,20,10,17.

Moving on to 'e': enallage and episome. Both start with 'e', so second letters are 'n' vs 'p'. 'en'
    comes before 'ep', so enallage (7) first, then episome (19).

Next is 'f': fraternity (6) and ft (14). 'fraternity' starts with 'fra', which comes before 'ft'
    because after 'f', 'r' comes before 't'. So fraternity (6), then ft (14).

Then 'g': greater_celandine (3), greco-roman_deity (4), ground_rule (30). Wait, all start with 'g'.

So need to sort these:

greater_celandine: g r e a t e r...

greco-roman_deity: g r e c o ...

ground_rule: g r o u n d ...

Wait, let me compare each one beyond the first letters.

All start with 'gr', so third letters are 'e' in greater and greco, but ground starts with 'gro'.

So:

Compare 'g r e a t e r...' vs 'g r e c o ...' vs 'g r o u n d...'

The first two have 'e' as the third letter after 'gr', which comes before 'o'. So greater_celandine and greco-roman_deity come before ground_rule.

Now between greater and greco:

Both start with 'gre'.

Next letters: in greater, it's 'a'; in greco, it's 'c'. Since 'a' (1st) comes before 'c' (3rd), so greater_celandine comes before greco-roman_deity.

So within 'g':

greater_celandine (3)

greco-roman_deity (4)

ground_rule (30)

Next is 'o': no 'o' at the beginning, except maybe I missed. Wait, we have 'p's?

Wait no, after 'g', moving on to 'h'? No, wait let me check all starting letters.

Looking back: a, b, c, e, f, g... then we have 'm', 'n', etc. So next would be 'm' and 'n'.

Wait no, perhaps I missed some letters. Wait the starting letters so far are:

amalgamator (a)

blowing_up (b)

blue_columbine (b)

Then c group.

e group.

f group.

g group.

What about 'h'? Probably none in this list.

Next letter after 'g' is 'm', then 'n'.

Looking at the data, do we have any words starting with 'm'? I don't think so. What about 'p'? No.

Wait wait, no: looking back:

After 'g's (3,4,30), what's next? Let's see all the other entries:

We had 1 to 30 as items, but perhaps some letters in between.

Looking at the list again, let me think of all words and their starting letters:

amalgamator: a

blowing_up: b

```
blue_columbine: b
```

```
capsicum_annuum_longum: c
```

```
carboniferous: c
```

```
cockeyed: c
```

```
compline: c
```

```
cyprinid_fish: c
```

```
enallage: e
```

```
episome: e
```

```
fraternity: f
```

```
ft: f
```

```
greater_celandine: g
```

```
greco-roman_deity: g
```

```
ground_rule: g
```

Then, what about 'h'? I don't think any words start with h.

Next letters after 'g' are 'h', then 'i', 'j', etc., but perhaps none in this list. So moving on to 'm'.

Wait, perhaps I'm overcomplicating. Maybe it's better to look at all the remaining entries and sort them by their starting letters.

Looking back:

The words after 'g' group are:

ground_rule (30)

Then, other words: ft is already in 'f' group.

Wait no, perhaps we have more.

What about 'm'? Let me see if any word starts with 'm'.

In the list given, I think not. Next letter would be 'n', but again, probably none except maybe 'none'. Not sure.

Then, letters like 'p': Do we have any words starting with 'p'? Yes: ft is f; what about others?

Wait no, perhaps 'p' comes after 'o', but in this list, I don't think so. So moving on to 'm' group, then 'n', etc.

Alternatively, maybe the next letters are 'h' through 'z', but since we have limited words, perhaps it's easier to proceed sequentially.

Wait, another approach: let me list all the remaining words after 'g's and sort them:

Looking at word IDs 1 to 30, I think the other words beyond what's already been sorted in a, b, c, e, f, g are:

We have 'm' perhaps? No. Wait, for example:

Wait, we have:

- System: (the initial problem statement, not part of the list)

Wait no, in the given data, I think all words after 'g's are:

ground_rule (30)

Then, looking through the initial list:

Wait, word 12 is missing? Or perhaps I miscounted.

Wait, let me recount the initial word numbers. The user provided a numbered list from 1 to 30. So we have to make sure all words are accounted for in the sort.

So far, after 'g's, which includes 3,4,30, what remains? Let me see:

From 1 to 30:

1: amalgamator (a)

2: ?

Wait no, wait perhaps I misread. Wait, let's look back at the original problem statement.

Wait the user wrote:

Please sort the following list of words alphabetically:

But then provided a list from 1 to 30 with some entries missing? Or is it a continuous list?

Wait actually, looking again: the initial list provided by the user in the problem seems incomplete or perhaps misrepresented. Let me recheck.

The original problem statement says:

Please sort the following list of words alphabetically:
1: amalgamator
2: blue_columbine
3: greater_celandine
4: greco-roman_deity
5: cyprinid_fish
6: fraternity
7: enallage
8: compline
9: blowing_up
10: capsicum_annuum_longum
11: carboniferous
12: cockeyed
13: ft
14: ground_rule

Wait, that's only up to 14. So perhaps the user intended a shorter list.

Wait no, in the problem statement above, it shows numbers from 1 to 30 with many entries. Wait
    maybe I'm confusing two different problems.

Wait wait, looking back:

The initial problem given is:

"Please sort the following list of words alphabetically:
1: amalgamator
2: blue_columbine
3: greater_celandine
4: greco-roman_deity
5: cyprinid_fish
6: fraternity
7: enallage
8: compline
9: blowing_up
10: capsicum_annuum_longum
11: carboniferous
12: cockeyed
13: ft
14: ground_rule"

Wait so it's 14 items. So the initial list is 1-14 as above.

So my mistake earlier was thinking of a longer list, but in reality, there are only 14 words to
    sort.

Therefore, after handling 'g' group, which includes:

3: greater_celandine

4: greco-roman_deity

14: ground_rule

Then the remaining letters start with letters beyond 'g'.

Looking at the remaining entries:

1: a
2: b
5: c
6: f
7: e
8: c
9: b
10: c
11: c
12: c
13: f
14: g

Wait no, 5 is cyprinid_fish (c), so it's in 'c' group.

So after processing all a, b, c, e, f, g groups, what remains? Let me list them:

a:1

b:2,9

c:5,8,10,11,12

e:7

f:6,13

g:3,4,14

Wait, so all 14 words are accounted for in these groups.

Therefore, the sorted order will start with a, then b's, then c's, e, f, g.

So after 'g' group (words starting with 'g'), there are no more letters left because we've covered up to 'g'. So the entire list is sorted as per these groups.

So compiling all this:

a:1

b:2,9

c:5,8,10,11,12

e:7

f:6,13

g:3,4,14

Wait no, the correct alphabetical order is a, b, c, e, f, g. Because after 'd' comes 'e', but we don't have any 'd's here.

So within each group:

a:1

b:2 (blue_columbine), 9 (blowing_up) because 'blowing_up' comes before 'blue_columbine'? Wait no, wait when sorting alphabetically, after the first letter, we look at subsequent letters. So ' blowing_up' starts with 'blow', and 'blue_columbine' starts with 'blue'. Since 'blow' comes before 'blue' because 'l' comes before 'u' in the third letter (after 'b'), so 'blowing_up' should come after 'blue_columbine'? Wait no: wait, let me think.

Wait when comparing two words starting with 'b', like 2: blue_columbine and 9: blowing_up:

The first letters are both 'b'. The second letters are 'l' in both. Third letters: for 2 it's 'u', for 9 it's 'o'. So since 'o' comes before 'u', 'blowing_up' (9) should come before ' blue_columbine' (2). Because 'blow' is before 'blue'.

Wait wait, no:

Wait the word 9 is blowing_up: 'b-l-o-w-i-n-g...' and word 2 is blue_columbine: 'b-l-u-e-...'

So after 'b', both have 'l'. Then next letters are 'o' vs. 'u'. Since 'o' comes before 'u' in the alphabet, 'blowing_up' (9) should come before 'blue_columbine' (2).

Therefore, within the 'b's, 9 comes first, then 2.

Similarly, for other groups:

c group: words are 5,8,10,11,12. Let me list them:

5: cyprinid_fish

8: compline

10: capsicum_annuum_longum

11: carboniferous

12: cockeyed

So let's order these alphabetically.

Compare the first letters after 'c':

compline (8) starts with 'co'

capsicum... (10): 'ca'

carboniferous (11): 'ca'

cockeyed (12): 'co'

cyprinid_fish (5): 'cy'

Wait no:

Let me write down each word's beginning:

5: cyprinid starts with 'c-y-p-r-i-n-i-d'... So the first letters after 'c' are 'y'.

8: compline 'c-o-m-p-l-i-n-e'

10: capsicum_annuum_longum 'c-a-p-s-i-c-u-m_...' so after 'c', it's 'a'.

11: carboniferous 'c-a-r-b-o-n-i-f-e-r-o-u-s' after 'c', 'a'.

12: cockeyed 'c-o-c-k-e-y-e-d'

So, among these:

The order should be based on the letters following 'c':

First, those starting with 'ca' (10 and 11), then 'co' (8 and 12), then 'cy' (5).

Within 'ca', compare next letters.

10: capsicum after 'ca', it's 'p'

11: carboniferous after 'ca', it's 'r'

So since 'p' comes before 'r', 10 comes before 11.

Then, within 'co':

8: compline and 12: cockeyed.

Looking at the third letter:

After 'c-o', for 8 it's 'm' (com...), for 12 it's 'c' (coc...). So 'c' comes before 'm'. Therefore, 12 comes before 8.

Wait wait no: let me think. Wait after 'co', the next letters are:

For 8: compline 'com'

For 12: cockeyed 'co' followed by another 'c'.

So the third letter is 'm' vs 'c'. But since in alphabetical order, 'c' comes before 'm', so 'coc' comes before 'com'.

Therefore, 12 (cockeyed) should come before 8 (compline).

Wait no: Wait, cockeyed starts with 'co', then another 'c', making it 'coc', whereas compline is 'com'. So after 'co', the next letters determine order.

Since 'c' comes before 'm', 'coc' < 'com', so cockeyed (12) comes before compline (8).

Wait no, because in dictionary order, each letter is compared one by one. Let's take it step by step:

Compare 12 and 8.

Both start with 'co'.

Then the third character: for 12 it's 'c' (as in 'cock'), and for 8 it's 'm' (as in 'com'). So since 'c' comes before 'm', cockeyed (12) should come before compline (8).

Wait, but wait, let me confirm with an example. For instance, between "coconut" and "comet", which comes first? "coconut" starts with 'coc', while "comet" starts with 'com'. Since 'c' < 'm' at the third position, "coconut" comes before "comet".

So yes, 12 (cockeyed) should come after 'ca' words and before 8.

Wait no: Wait 10 is 'capsicum...', which starts with 'ca', so it's first. Then 11: 'carboniferous' also starts with 'ca', but since next letters are 'p' vs 'r', 10 comes before 11.

Then, after 'ca' group, we have the 'co' words: 12 (cockeyed) and 8 (compline). As per above, 12 comes before 8 because 'coc' < 'com'.

After that, we have 5: cyprinid_fish, which starts with 'cy', so it's last in the 'c' group.

So within 'c' group:

10,11,12,8,5? Wait no. Let me clarify.

Wait, the 'c' group words are 5,8,10,11,12.

We have to order them alphabetically.

Starting with the first letters after 'c':

10: capsicum 'ca'

11: carboniferous 'ca'

12: cockeyed 'co'

8: compline 'co'

5: cyprinid_fish 'cy'

So, within 'c', order is determined by the subsequent letters.

First, all words starting with 'ca' (10 and 11) come before those starting with 'co' (12 and 8), which in turn come before 'cy' (5).

Within 'ca':

Compare 10 and 11:

After 'ca', next letter for 10 is 'p' (from capsicum ca-p-...), and for 11 it's 'r' (carboniferous ca-r-...). Since 'p' comes before 'r', 10 comes before 11.

Then, within 'co':

Compare 12 and 8:

After 'co', next letters are 'c' for 12 (cock-) and 'm' for 8 (com-). So 'c' < 'm', so 12 comes before 8.

So the order of 'c' group is: 10,11,12,8,5.

Wait no. Wait:

Wait, 10 is capsicum_annuum_longum starts with 'ca'

Then 11: carboniferous also 'ca', but since after 'ca', 10 has 'p' and 11 has 'r', so 10 before 11.

Next are the 'co' words. Between 12 (cockeyed) and 8 (compline): as discussed, 12 comes before 8 because 'coc' < 'com'.

So after 11, we have 12, then 8.

Finally, 5: cyprinid_fish is last in the group.

So within 'c' group order is:

10,11,12,8,5.

Now moving on to the other groups.

After all that, let's put it all together step by step.

The overall categories are:

Group A: Words starting with letters before 'a'

Wait no. Let me think again. Wait, this might be getting complicated, but I'm trying to structure it properly.

Alternatively, perhaps it's easier to list out the words in order, comparing each as needed.

But maybe a better approach is to note that the initial system message seems incomplete or cut off. The user probably pasted an incomplete message, ending with "### System: Please reason step by step, and put your final answer within \boxed{}."

So perhaps I should provide guidance on how to alphabetize a list of words based on their prefixes.

Wait, but the problem isn't clear because the system message seems truncated. However, from the context, it appears that the user might have intended to ask about the alphabetical order of certain terms, perhaps similar to "capsicum", "carboniferous", "cockeyed", "compline", and "cyprinid_fish".

If I were to arrange these in alphabetical order, here's how I would do it:

1. Start by comparing each word letter by letter.

2. The words are: capsicum, carboniferous, cockeyed, compline, cyprinid_fish.

3. All start with 'c', so move to the second letter.

4. For capsicum and carboniferous, the second letter is 'a'.

5. For cockeyed and compline, it's 'o'.

6. Cyprinid starts with 'y' after 'c'.

So first group: 'ca' words capsicum and carboniferous.

Between these two:

After 'ca', next letters are 'p' in capsicum and 'r' in carboniferous. Since 'p' comes before 'r', capsicum comes before carboniferous.

Next group: 'co' words cockeyed and compline.

Compare the third letter after 'co':

- Cockeyed has 'c' (as in 'coc'), while compline has 'm'.

So 'coc' comes before 'com', meaning cockeyed comes before compline.

Finally, cyprinid_fish starts with 'cy', which is after 'co', so it's last.

Putting it all together:

1. capsicum
2. carboniferous
3. cockeyed
4. compline
5. cyprinid_fish

So the order would be: capsicum, carboniferous, cockeyed, compline, cyprinid_fish.

Wait no, because I think I made a mistake with 'cock' vs 'com'.

Let me confirm using actual dictionary ordering:

- Compare "carboniferous" and "capsicum": since 'ca' is same, next letters are 'p' in capsicum vs 'r' in carboniferous. So capsicum comes before carboniferous.

```
Now for the 'co' group: cockeyed vs compline.

Compare each letter after 'c':

- Both have 'o' as second letter.

Third letters: 'c' in cockeyed (making it 'coc') and 'm' in compline ('com'). Since 'c' comes
    before 'm', cockeyed should come before compline.

So the correct order is:

1. capsicum
2. carboniferous
3. cockeyed
4. compline
5. cyprinid_fish

Therefore, if these words were in a list, they would be ordered as above.
</think>

The alphabetical ordering of the given words is determined by comparing each letter sequentially.
    The correct order is:

1. capsicum
2. carboniferous
3. cockeyed
4. compline
5. cyprinid_fish

**Step-by-Step Explanation:**

1. **Compare Initial Letters:** All start with 'c', so proceed to the next letter.
2. **Second Letter Comparison:**
   - "capsicum" and "carboniferous" have 'a'.
   - "cockeyed" and "compline" have 'o'.
   - "cyprinid_fish" has 'y'.
3. **Order by Second Letter:** 'a' comes before 'o', which comes before 'y'. So, "capsicum" and "
   carboniferous" come first, followed by the 'co' group, then "cyprinid_fish".
4. **Within 'ca' Group:**
   - Compare next letters after 'ca': 'p' (capsicum) vs 'r' (carboniferous). 'p' comes before 'r',
        so capsicum precedes carboniferous.
5. **Within 'co' Group:**
   - After 'co', compare the third letter: 'c' in cockeyed vs 'm' in compline. 'c' comes before 'm
        ', so cockeyed comes before compline.

**Final Order:**

\boxed{capsicum, carboniferous, cockeyed, compline, cyprinid\_fish}
```

### A.7 Example of overthinking by o3-mini

Below is an example of overthinking by o3-mini for the first list of length 256 of the English task. We note that we do not have access to the actual reasoning tokens and, therefore, only report the summary available on the Web-frontend. Therefore, it is likely that the actual reasoning was somewhat different, including possibly longer and without the artifact of the language change that we observe.

```
Okay, I need to sort the list according to Pythons standard method, which is alphabetically (
    lexicographically) by default using string comparison. I think this means well use the sorted()
     function for that.
```

The sorting in Python uses Unicode codepoint ordering, which sorts lowercase letters, numbers, and underscores in ascending order.

Ill produce the sorted list now and ensure I output it in exactly the same format but sorted. Lets compute it!

Alright, I need to emulate the sorted() function on this list. I'll sort it lexicographically, which means following alphabetic rules.

Im going to mentally process this, considering things like "42nd" versus "aerolite", "afterdamp", "antitoxin", etc. and ensuring proper sorting with underscores and numbers taken into account.

Let's just sort them manually here: Well handle entries like "awkward", "barograph", and "cheating", and see how they stack up.

Okay, I think I have the sorted order! Let's compute the result now.

Okay, here's the list of elements I need to sort.

Im going to go over them and make sure Ive listed everything correctly. I see some familiar ones like "little_giant", "air_mail", "aerolite," and "intelligibility."

Hey, theres a duplicate of "chongqing" that I spotted. I need to confirm if the result will have both or eliminate one.

Let's proceed with sorting the list properly and accurately. Ill make sure I get all the details right.

Okay, now I see I need to sort this gigantic list alphabetically. Ill take care to follow the correct lexicographical order: numbers first, then letters. For example, "42nd" should come before words starting with 'a'.

Then, Ill focus on sorting those that start with 'a' based on the second letter, like sorting "acoustic_modem" before "afterdamp" because 'c' (99) comes before 'f' (102).

Ill be sure to check everything carefully to avoid mistakes!

Ich werde die Wörter nach ihren Anfangsbuchstaben sortieren: für helicopter und healthier vergleiche ich die Buchstaben und komme zu dem Schluss, dass healthier vor helicopter kommt. Weiterhin vergleiche ich helicopter mit healthcare_delivery und bestimme die Reihenfolge basierend auf den Buchstaben. Auch für die folgende Gruppe wie intelligibility und independency ordne ich nach den Buchstaben ab der dritten Stelle und sortiere so die Begriffe. Eventually, I get a final sorted order for the terms.

Ich vergleiche die Begriffe, die mit 's' beginnen. Zwischen 'sapota' und 'saddle_blanket': 'saddle_blanket' kommt zuerst, weil d (100) < p (112). Bei der Sortierung der Begriffe 'simon_marks', 'sir_james_matthew_barrie', und 'silver' ordne ich sie zu, da 'simon_marks' < 'sir_james_matthew_barrie' und 'sir_james_matthew_barrie' < 'silver'. Weiter vergleiche ich 'statue_of_liberty', 'steel_company' und 'sterilizer', wobei 'statue_of_liberty' vor 'steel_company' kommt. Schlielich ordne ich 'subsist' und 'sulphur_dioxide'.

Ich habe die Wörter der Gruppen 's', 't', 'u', 'v', 'w', und 'y' sortiert. Die Reihenfolge für 's' ist jetzt: 'saddle_blanket', 'sapota', 'scentless_hayweed', 'shaman', 'silver', und so weiter. Gleiches gilt für die anderen Buchstaben von 't' bis 'y'. Es gibt nun eine vollständige, alphabetisch sortierte Liste, beginnend mit '42nd' gefolgt von A-Wörtern und allen anderen Gruppen. Die endgültige Liste enthält alle richtig sortierten Begriffe und ist nun bereit zur Präsentation.

I see that "private_treaty" belongs in the p group, right after "profitably," since "private_treaty" comes alphabetically before it. I adjust the p group order. Similarly, "reflex_response" and "rodomontade" need to be added to the r group, and sorting them places "reflex_response" before "reprise." Also, in the w group, "white_sheep" is added after "werner_karl_heisenberg," and I also need to adjust "chongqing" in the c group, following "chicken_purloo." It's a bit of a puzzle!

It looks like "impure" fits in the i group between "input_device" and "intension." I'll update the rest of the group accordingly. The s group gets a new member, "scheldt_river," which belongs after "scentless_hayweed" but before "shaman," so Ill reorder. The g group now includes "gnaphalium" between "genus_garrulus" and "golden_oriole." As for "redheaded_woodpecker," it'll be placed after "reflex_response" in the r group. And "nutritionist" fits nicely in the n group!

