# OpenReview forum: "SortBench: Benchmarking LLMs based on their ability to sort lists"
_TMLR — Rejected by TMLR_

### Review · Reviewer_Kx4i · 2025-05-09

**Summary Of Contributions:**

The authors propose a benchmark designed to assess both the general sorting capabilities of LLMs and also to understand which properties of data make sorting more difficult for them. They also apply this benchmark to seven state-of-the-art LLMs, including current test-time reasoning models, and extract interesting insights from the results obtained.

**Audience:**

Yes

**Broader Impact Concerns:**

No concerns regarding ethical implications.

**Claims And Evidence:**

Yes

**Requested Changes:**

I think it would be positive to include as an appendix some concrete real examples of prompts provided to the LLMs during experiments. This, together with what is indicated in section 2.1, I believe would help to more clearly illustrate the type of input provided to the LLM in each experiment.

One of the premises of this work is to study, analyze, and evaluate the limited capabilities of LLMs when it comes to sorting lists. However, if one looks at Table 1, overall, such clear limitations do not appear to exist, given that almost all models provide reasonably high values for the metrics. In fact, if my calculations are correct, across all tasks (last 7 rows of the Table), LLMs offer an average ModelScore of approximately $0.8$ (with o3-mini yielding an average ModelScore over all tasks of $0.977$). Can the authors comment on this?

In page 7, what do the authors mean by _"We ran the NumberSorting two times"_? What is this "NumberSorting"? If I'm not mistaken, this is the first time this expression ("NumberSorting") appears in the paper.

I think it'd be interesting to include average response time for each model and problem/task. This would help to quantify statements like _"For one of the examples we tried, i.e., the basic English task, the reasoning process was very long"_.

My advice to authors would be to use a slightly more objective and neutral language, free from adjectives and subjective judgments, such as when they state that _"The clear disappointment in our experiment was the DeepSeek-r1 model."_

I would try to enlarge Figure 2 so that it could be read better.

Typos:
* page 1: "requiring an lexicographic"
* page 4: "items needed the be cast"
* page 5: I'd try to be consistent with the notation. Sometimes you use "DeepSeek-r1-70b" and sometimes you use "DeepSeek-r1-70B"
* page 7: "outpeforms the other models"
* page 7: "they require a comparably many tokens for the list items"
* page 8: "similar strengths and weakness"
* page 9: "the the second"

**Strengths And Weaknesses:**

Strengths:
* The paper addresses an important topic, given that understanding and analyzing the capabilities of LLMs is highly relevant to advancing the field.
* I believe the empirical evaluation is quite solid: it employs tasks of increasing difficulty and nature, assesses the impact of list length, and uses metrics that I believe are relevant. The selected LLMs also seem sufficient in number and diversity.

Weaknesses:
* The work, in my opinion, does not present major weaknesses or limitations, perhaps the greatest of which is its very specific nature: it focuses solely on sorting lists without paying attention to other tasks of potential interest. So the scope of the work could be said to be somewhat narrow/limited.
* If the main contribution of this work is creating the benchmark (SortBench v1.0), I guess the authors should share it (either the exact prompts, the code to generate them, etc.). I may be mistaken, but I didn't find any link to the benchmark so the scientific community could evaluate their LLMs on it.

---

> ### Author Response · Authors · 2025-05-23
> **Response to Reviews**
>
> Thanks for your review. We tried to address everything. We uploaded an updated version of the manuscript that addresses all changes. Further, you can access an annotated version of the manuscript with changes (except typo corrections) highlighted in blue here: https://files.catbox.moe/f7us43.pdf
>
> **Weaknesses:**
>
> > The work, in my opinion, does not present major weaknesses or limitations, perhaps the greatest of which is its very specific nature: it focuses solely on sorting lists without paying attention to other tasks of potential interest. So the scope of the work could be said to be somewhat narrow/limited.
>
> Regarding the general critique of the usefulness, we refer to the Limitations (Section 4.4), which we futher updated based on reviewer comments to address this question.
>
> > If the main contribution of this work is creating the benchmark (SortBench v1.0), I guess the authors should share it (either the exact prompts, the code to generate them, etc.). I may be mistaken, but I didn't find any link to the benchmark so the scientific community could evaluate their LLMs on it.
>
> The manuscript currently contains a blinded reference in Section 3. We will add a link to GitHub with all code and data for easy re-use and extension by others to a possible unblinded, camera-ready version.
>
> **Requested Changes:**
>
> > I think it would be positive to include as an appendix some concrete real examples of prompts provided to the LLMs during experiments. This, together with what is indicated in section 2.1, I believe would help to more clearly illustrate the type of input provided to the LLM in each experiment.
>
> We extended the benchmark with examples for each input task (Appendix A.5) as well as one example of the overthinking process to also exemplify what can happen with the outputs (Appendix A.6 and A.7).
>
> > One of the premises of this work is to study, analyze, and evaluate the limited capabilities of LLMs when it comes to sorting lists. However, if one looks at Table 1, overall, such clear limitations do not appear to exist, given that almost all models provide reasonably high values for the metrics. In fact, if my calculations are correct, across all tasks (last 7 rows of the Table), LLMs offer an average ModelScore of approximately 0.8 (with o3-mini yielding an average ModelScore over all tasks of 0.977). Can the authors comment on this?
>
> This depends on the tasks. For example, for 03-mini  we also, indeed, come to the conclusion that this model *almost* solves the sorting task in Section 3.1. However, the model still has two problems: With the NumberWords tasks, where it often converts the list to integers and then sorts them, as well as just not changing list items for long floats with many digits after the comma and uncommon strings (i.e., not just English words). So yes, average performance across all tasks (incl. simpler ones) is very strong, but the even the best models still struggle a bit with the more complex tasks.
>
>
> > In page 7, what do the authors mean by "We ran the NumberSorting two times"? What is this "NumberSorting"? If I'm not mistaken, this is the first time this expression ("NumberSorting") appears in the paper.
>
> This was a typo. We meant the NumberWords tasks. This is now fixed.
>
> > I think it'd be interesting to include average response time for each model and problem/task. This would help to quantify statements like "For one of the examples we tried, i.e., the basic English task, the reasoning process was very long".
>
> We hesitate to put numbers to the actual response times, as this is also a factor of the underlying hardware and optimization of the performance. For example, GPT-4o has lower response time than our locally hosted Llama-3.1-70b model. We also avoid the reporting statistical data about the reasoning length by o3-mini, since we do not have access to the reasoning tokens, we only see a summary of the reasoning process. That being said, we agree that more data regarding this would be nice. Since we believe that report of charts with trends for all tasks might be misleading, since we do not know how much actually happened in the background, we rather opt for an anectdotal approach here and report the concrete example in Appendix A.7.
>
> > My advice to authors would be to use a slightly more objective and neutral language, free from adjectives and subjective judgments, such as when they state that "The clear disappointment in our experiment was the DeepSeek-r1 model."
>
> We phrased this more neutrally: "Our expectation of strong performance for test-time reasoning models was not fulfilled by the DeepSeek-r1 model."
>
> > I would try to enlarge Figure 2 so that it could be read better.
>
> We increased the size of the figure to the full column width so that the text should now be easier to read.
>
> > Typos: [...]
>
> Thanks! All fixed.

---

> > ### Comment · Reviewer_Kx4i · 2025-05-24
> >
> > Dear all,
> >
> > I've read all authors' responses to my comments, and I'm happy with them. According to the TMLR acceptance criteria (https://jmlr.org/tmlr/acceptance-criteria.html), acceptance of a submission should be based on positive answers to the following two questions:
> > - _Are the claims made in the submission supported by accurate, convincing and clear evidence?_
> > - _Would some individuals in TMLR's audience be interested in the findings of this paper?_
> >
> > From my point of view, the answer to both questions is affirmative so far; therefore, I lean toward accepting the paper.
> >
> > In any case, I recommend that the authors carefully review the writing. For example, based on the modifications made to the paper, I would suggest the following changes (among others):
> > - "This allows us deeper insights into the differences between models and their ability to produce outputs that fulfill the desired properties." &rarr; "This allows us to gain deeper insights into the differences between models and their ability to produce outputs that fulfill the desired properties."
> > - "the GPT-4o model was also most stable towards longer sequences" &rarr; "the GPT-4o model was also the most stable with respect to longer sequences"
> > - "This indicates that achieving stronger performance at long contexts is possible as is demonstrated by OpenAI’s models." &rarr; "This indicates that achieving stronger performance at long contexts is possible as demonstrated by OpenAI’s models."
> > - "we cannot answer due the lack of openness of OpenAI models." &rarr; "we cannot answer due to the lack of openness of OpenAI models."
> > - "This also helps us to diagnose is the length of the context itself becomes a problem" &rarr; "This also helps us diagnose whether the length of the context itself becomes a problem."
> > - "all models are able to provide fully sorted integer lists for all length." &rarr; "all models are able to provide fully sorted integer lists for all lengths."
> > - "with English words we discuss above" &rarr; "with English words we discussed above"
> > - "We also observe the same problems with validity we have for unsorted list." &rarr; "We also observe the same problems with validity we have for unsorted lists."
> > - "Overall, this debug tasks confirms that the problems we found with sorting a prevalent even in a simpler context and just further exacerbated by more difficult sorting tasks." &rarr; "Overall, these debug tasks confirm that the problems we found with sorting are prevalent even in a simpler context and are just further exacerbated by more difficult sorting tasks."
> > - "The second debug tasks, i.e., providing each list item twice, helps us to understand if such duplication negatively affect how the LLMs work." &rarr; "The second debug task, i.e., providing each list item twice, helps us understand if such duplication negatively affects how the LLMs work."
> > - "The exception are the very large model models (GPT-4o, o3-mini, and Claude-3.5-Sonnet)." &rarr; "The exception are the very large models (GPT-4o, o3-mini, and Claude-3.5-Sonnet)."
> > - "However, even these models have problems with the faithfulness for the English words, Claude-3.5 for Floating point numbers." &rarr; "However, even these models have problems with faithfulness for English words; Claude-3.5 also struggles with floating point numbers."
> > - "This give us a good indication about what is required to allow models to disambiguate two equal items: are very large number of parameters" &rarr; "This gives us a good indication of what is required to allow models to disambiguate two equal items: a very large number of parameters"
> > - "However, while the task we study is only sorting, the model capabilities we study also include the ability to be faithful the task description" &rarr; "However, while the task we study is only sorting, the model capabilities we study also include the ability to be faithful to the task description,"
> >
> > Best,

---

> > > ### Author Response · Authors · 2025-05-26
> > >
> > > Thank you both the for the positive feedback and the list of issues we can address to improve the writing! We will also do another round of internal proof reading to find and fix additional mistakes and update the paper later this week.

---

### Review · Reviewer_NGZ6 · 2025-05-11

**Summary Of Contributions:**

This paper proposes SortBench to test LLMs with sorting problems in different difficulty levels. The evaluation reveals insights like test-time reasoning has a tendency to overthink problems and current models are still struggling with long-context problems.

**Audience:**

Yes

**Broader Impact Concerns:**

I didn't see any broader impact concern around this work.

**Claims And Evidence:**

Yes

**Requested Changes:**

High-level Suggestions:
1. I would suggest removing redundant metrics and providing justification for metrics design.
2. The paper can be improved by conducting analysis on the correlation between SortBench performance and other benchmarks people generally care about. If improvement on SortBench could potentially lead to improvement of LLMs' core capabilities, SortBench would have more value for model developers.

Low-level Suggestions:
1. For Table 1, it is better to keep the order of methods the same in different subset for easy comparison. If possible, the authors could consider moving "Basic", "Advanced", "Debug", "All tasks" to the column to have a large table with each method occupying a single line. This would make the table more compact and easy-to-compare.
2. Some wordings in the paper do not read scientific. For example, the heading of Section 3.3.
3. Some content in the main paper are not very important and I would suggest a cut. For example, the detailed analysis of format error (Figure 2) could probably go into Appendix.

**Strengths And Weaknesses:**

Strengths:
1. SortBench is a synthetic dataset that can provide thorough evaluation of LLM's sorting capability. The dataset itself is well-designed with samples covering different levels.
2. The paper conducts thorough experiments and analysis, probing different aspects of LLMs' capabilities (e.g., reasoning, test-time scaling, long context).

Weaknesses:
1. Given SortBench is a synthetic dataset and we do not need to rely on LLMs to conduct sorting, it is unclear whether the insights obtained through SortBench evaluation can be generalized to LLMs' capabilities on other tasks that people care about. For example, does the results on long lists correlate to LLMs capability on long-context benchmarks? Does the results on SortBench correlate to LLMs' performance on other reasoning benchmarks?
2. The metrics defined in this work are complicated and not well-justified. For sorting correctness, I am unsure about whether the current definition of SortingScore is better than simply using a binary metric since the sorting task fails as long as there is at least one unordered pair. Similarly, for FaithfulnessScore, it is unclear why it computes an average of the two error types. The total score is merely a tranformation of other scores, which does not seem to be necessary either.
3. The paper proposes the Debug subset to help probe LLMs' capabilities for a deeper understanding. While this attempt is laudable, the insights from the Debug subset are unclear, especially given that the ranking of different models on this subset is different from other subsets. How shall we use the Debug subset to improve LLMs?

---

> ### Author Response · Authors · 2025-05-23
> **Response to Reviews**
>
> Thanks for your review. We tried to address everything. We uploaded an updated version of the manuscript that addresses all changes. Further, you can access an annotated version of the manuscript with changes (except typo corrections) highlighted in blue here: https://files.catbox.moe/f7us43.pdf
>
> **Weaknesses:**
>
> > Given SortBench is a synthetic dataset and we do not need to rely on LLMs to conduct sorting [...]
>
> Regarding long contexts, we refer to our answer below to the requested changes.
>
> Regarding the general critique of the usefulness, we refer to the Limitations (Section 4.4), which we further updated based on reviewer comments to address this question.
>
> > The metrics defined in this work are complicated and not well-justified. [...]
>
> From our perspective, using a simple binary score would oversimplify the problem and make it harder to judge differences between models and measure progress towards solving this task. We extended Section 2.4 with an explanation for this choice based on the ability to gradually measure progress.
>
> We admit that there would be other ways to design these metrics, including not taking the average of the local and global perspective of sorting into account or distinguishing in a categorial manner for the different issues with the validity. However, we would argue that this additional complexity allows us to have more reliable measurements regarding model capabilities.
>
> As for the total score, this is indeed only a combination of the other metrics. However, this is by design to have what is typically desired from benchmarks at the end of the day: having a single number that can be used to rank different models. In our case, this requires taking the different properties into account, which we do through this combination.
>
> > The paper proposes the Debug subset to help probe LLMs' capabilities for a deeper understanding. [...] How shall we use the Debug subset to improve LLMs?
>
> Indeed, we neglected discussing the utility of the debug task in the results. We now added a new Section 3.8 with insights from the debug tasks.
>
> **Requested Changes:**
>
> > I would suggest removing redundant metrics and providing justification for metrics design.
>
> See above.
>
> > The paper can be improved by conducting analysis on the correlation between SortBench performance and other benchmarks people generally care about. [...]
>
> Thanks for this suggestion! We looked into recent work on long-contexts and were particularly impressed by a recent paper from Modaresi et. al. (https://arxiv.org/pdf/2502.05167?). Notably, our results indicate the same trends at a fraction of the compute budget required and with the use of synthetic, hard to overfit data. We extended our discussion of the long-context issues in Section 3.6 to put our results into this recent context and also highlight the advantage of SortBench regarding compute. We do not separately highlight the advantage of avoiding overfitting, as this is already discussed in Section 4.3.
>
> > For Table 1, it is better to keep the order of methods the same in different subset for easy comparison. [...]
>
> We fixed the order of the models as suggested, based on the performance over all tasks.
>
> We also tried the reformatting as suggested. However, at the current precision of reporting (three digits after the comma), this is not possible with the width we have available for TMLR. We could use a smaller font-size or reduce the reporting precision, but that would just add different readability problems, resp. reduce the accuracy of the results. Due to that, we stay in the current format.
>
> > Some wordings in the paper do not read scientific. For example, the heading of Section 3.3.
>
> We understand that long, informative headings are often avoided in computer science papers, where headings often follow generic structural names and are often designed to be very short. However, in science in general, longer titles that try to summarize key aspects of the subsequent section are not uncommon. Since we believe that such titles actually aid readers see valuable information fast and, possibly, attract their attention to the sections that would interest them most, we did not modify this.
>
> > Some content in the main paper are not very important and I would suggest a cut. For example, the detailed analysis of format error (Figure 2) could probably go into Appendix.
>
> We believe that Figure 2 has value for improving training procedures of LLMs, as this highlights issues that could be addressed by providing appropriate fine-tuning data that specifically avoids these issues. Consequently, we decided to keep this in the main body of the paper for now, but would be open for moving this to the Appendix if this is seen as a blocking issue for acceptance.

---

### Review · Reviewer_1Fzq · 2025-05-18

**Summary Of Contributions:**

The paper proposes a benchmark based on a synthetic sorting problem. The paper found that o3-mini dominates the performance of the three tasks the author proposed. Author found all model has the issue of faithfulness: missing or adding number to the original to-be-sorted list.

**Audience:**

No

**Claims And Evidence:**

No

**Requested Changes:**

See weakness above, mainly point 2: span beyond just sorting question, and point 3,4: more in-depth experiment analysis to draw more reliable conclusion.

**Strengths And Weaknesses:**

Strength:
The paper is easy to understand. Researching to faithfulness seems interesting

Weakness
1. What is "test-time reasoning", Correspondingly, there should be "training-time reasoning"? This is a very odd word to use, instead you should introduce "o1-like" models or "large reasoning models" at the beginning of introduction. Please don't use terminology without defining.
2. "Sorting task" is a very specific type of problem. While it's nice to use it as a proxy for accessing reasoning capabilities, relying on only one type of task is extremely limited. E.g. To access neural algorithm reasoning, CLRS-30 [1] spans it's benchmark from 30 classic algorithms. I think a benchmark, aimed to evaluate llm's reasoning capabilities like this, should go beyond one type of task to reach to any reliable results
3.  The paper draws the conclusion that "test-time (scaling for) reasoning has a tendency to overthink problems, which leads to performance degradation". This is from the observation that the r1 model performs poorly in the benchmark. However, o3-mini is also a long-term reasoning model, but its performance dominates the benchmark.  Btw, this phenomenon has been systematically studied at [2]

4. There are quite a lot of missing details, and there is no in-depth analysis of the model's performance, and t. E.g. what's the temperature for each model, is the metric average of pass@1 or pass@1 when set temperature to 0. Can we have a evaluation on pass@N to see how good the model's potential is for solving the problem? Examples of the questions and input prompt? Examples of failure cases?

[1] Veličković, Petar, et al. "The clrs algorithmic reasoning benchmark." International Conference on Machine Learning. PMLR, 2022.
[2] Chen, Xingyu, et al. "Do not think that much for 2+ 3=? on the overthinking of o1-like llms." arXiv preprint arXiv:2412.21187 (2024).

---

> ### Author Response · Authors · 2025-05-23
> **Response to Reviews**
>
> Thanks for your review. We tried to address everything. We uploaded an updated version of the manuscript that addresses all changes. Further, you can access an annotated version of the manuscript with changes (except typo corrections) highlighted in blue here: https://files.catbox.moe/f7us43.pdf
>
> **Weaknesses:**
>
> > What is "test-time reasoning", Correspondingly, there should be "training-time reasoning"? [...] Please don't use terminology without defining.
>
> While we are also not fully happy with this term, it is starting to become common to describe the technique used by these models. We added the following clarification when we first use the term in our paper: "i.e., a variant of test-time compute efforts used for reasoning and planning [(Ji et al., 2025)](https://arxiv.org/abs/2501.02497)".
>
> > "Sorting task" is a very specific type of problem. [...] I think a benchmark, aimed to evaluate llm's reasoning capabilities like this, should go beyond one type of task to reach to any reliable results
>
> We differentiate our work from other benchmarks now as following in the introduction.
>
> Please note that we to CLRS-text instead of CLRS-30, as this is a newer variant of the benchmark that uses textual representations directly instead of graph structures.
>
> Additionally, we would argue that studying single algorithmic tasks also has value, when done in sufficient depth. Our experiments are designed to understand three things: do embeddings contain the information required for sorting (algorithmic task), can LLMs reliably follow such a simple instruction (faithfulness), and the impact of the length and type of list on these properties. We also limit our discussion in Section 4.1 to these aspects: sorting seems to generally work, though the models struggle with corner cases. Test-time reasoning is in general helpful for this, but can also lead to performance degradations due to overthinking.
>
> Nevertheless, we acknowledge that ideally our approach would be used for more algorithmic tasks already, but this requires additional design of suitable metrics (e.g., for pattern matching), handling of corner cases for this task for the validity of outputs, etc., leading to a fairly large amount of additional effort. Instead, we extended the conclusion with a statement that suggest this as opportunity for future work.
>
> > The paper draws the conclusion that "test-time (scaling for) reasoning has a tendency to overthink problems, which leads to performance degradation". [...] However, o3-mini is also a long-term reasoning model, but its performance dominates the benchmark.
>
> We discuss this in a more fine-grained manner in Section 3.1 (o3-mini) and 3.5 (DeepSeek-r1), where we do not make such a broad statement. While o3 has the best performance, it also had problems with overthinking, that we discuss in Section 3.2. Nevertheless, we agree that the conclusion right now is too negative and changed the discussion there to be more balanced.
>
> > Btw, this phenomenon has been systematically studied at [2]
>
> We added the following regarding Chen et al.'s [2] work to Section 3.1, i.e., the first time we discuss overthinking in the paper.
>
> > There are quite a lot of missing details [...]
>
> We added information regarding how we ran the experiment as new Appendix A.3, incl. repetitions, hardware used, and temperature.
>
> An evaluation of pass@N is not feasible due to the computational effort. The local experiments with multiple 70B models were already consuming our available resources for a notable amount of time. While we do not have N repetitions for each list, we have 10 different lists for each task, which also accounts for randomness, just with respect to different inputs with the same structural properties (list type and length). Confidence intervals averaged over all tasks for all lengths are reported in Appendix A.2 together with other statistical properties.
>
> Since we used the default temperature of 1 for most of our experiments, we ran a new experiment to understand the impact of this choice in comparison to a temperature of 0. We selected GPT-4o-mini for this experiment, as this a computationally efficient option that performs quite well in our benchmark. The results of this experiment are reported in the Appendix A.4. In summary, there is no big difference between using a temperature of 1 or a temperature of 0. We believe this is the case because neither the order nor the faithfulness are probabilistic properties, meaning most probabilities are very close to one, limiting the impact of the temperature.
>
> If required, we would also run this experiment for the other models, though this would take a couple of weeks for the locally hosted models (mostly because of the very large number of thinking tokens from DeepSeek-r1) and cost several hundred dollars for the OpenAI and Anthropic models, for what we believe would not be a lot of added value to our experiment results.
>
> **Requested Changes:**
>
> See above.

---

> > ### Comment · Reviewer_1Fzq · 2025-06-07
> >
> > While I appreciate the author's response, my major concerns are not resolved.
> >
> > * Evaluating on a single reasoning task, despite having a comprehensive design, is still limited because the reasoning mode is restricted. As an evaluation benchmark paper, the conclusion raised from such a design can be biased.
> > * Following the previous point, the overthinking problem the author claimed is vaguely explained and clarified, with no supporting quantitative or qualitative study.
> > * While I agree that api calling can be costly, I disagree with the author that additional experiments are not feasible. One do not need to use a 70b model to provide evidence in the experiments section. One can always use 7b/3b/1.5b/0.5b models from Qwen/Deepseek/Llamma/Phi, these models can be run on as low as 24GB GPUs and there is no reason not to provide these important experiment results.
> >
> > In all, I lean towards Reject / Major revision.
> >
> > Best

---

> > > ### Author Response · Authors · 2025-06-23
> > >
> > > First of all, sorry for the slow response.
> > >
> > > As discussed above, we disagree regarding the merits of studying single issues in depth.
> > >
> > > As for the overthinking, we wonder why the reviewer argues that there is no supporting evidence. We did study this qualitatively by looking at the results and even provide an example for Overthinking with DeepSeek in Appendix A.6. We also provide quantitative evidence through the relationship between the number of thinking tokens and the validity score, directly showing that longer though processes of the DeepSeek-r1 model led to invalid lists.
> > >
> > > Finally, we agree that we could easily conduct more experiments with smaller models. The question is what the additional insights through this would be. Our results already, fairly conclusively, demonstrate that sorting capabilities get stronger with larger models. Thus, while including smaller models could help us better understand lower-bound performance, we do not see other upsides, and, importantly, do not believe that this could strengthen any of the conclusions we draw that apply to the larger models.
> > >
> > > Consequently, while we understand your points, we disagree with the notion that they negatively affect the soundness of our work. They might affect external validity (generalization to other tasks, smaller models), but we believe that all claims we make are sufficiently substantiated.

---

### Author Response · Authors · 2025-05-23
**Response to Reviews**

Thanks to all reviewers for your valuable comments and insights into our work. We responded to all reviews and uploaded and updated version of our manuscript. You can also access a version with changes highlighted in blue here: https://files.catbox.moe/f7us43.pdf

We look forward to discuss additional issues or your recommendation.

---

### Decision · Action_Editor_TYgS · 2025-06-27

**Recommendation:** Reject

**Audience:**

Yes

**Audience Explanation:**

The benchmark would potentially be of interest to some in the TMLR audience, who might want to give a try of their model on SortBench. But given the limitations of the benchmark, it is unclear how many will be interested.

**Claims And Evidence:**

No

**Claims Explanation:**

While the new benchmark (SortBench) is tested on multiple LLMs, reviewers remained unconvinced by several key claims, despite further discussions and revision of the paper. One is that the benchmark is too limited (consisting of 1 type of tasks), to provide meaningful insights about model capabilities beyond what is already offered by existing benchmarks. Also, the accuracy metrics are already high, suggesting a risk of satiation soon and limited power of model evaluation. Finally, some of the findings are not sufficiently supported, such as that overthinking degrades performance, but the editor feels this is relative minor compared to the previous limitations.